# Inherited C-terminal TREX1 variants disrupt homology-directed repair to cause senescence and DNA damage phenotypes in *Drosophila*, mice, and humans

Age-related microangiopathy, also known as small vessel disease (SVD), causes damage to the brain, retina, liver, and kidney. Based on the DNA damage theory of aging, we reasoned that genomic instability may underlie an SVD caused by dominant C-terminal variants in TREX1, the most abundant 3′−5′ DNA exonuclease in mammals. C-terminal TREX1 variants cause an adult-onset SVD known as retinal vasculopathy with cerebral leukoencephalopathy (RVCL or RVCL-S). In RVCL, an aberrant, C-terminally truncated TREX1 mislocalizes to the nucleus due to deletion of its ER-anchoring domain. Since RVCL pathology mimics that of radiation injury, we reasoned that nuclear TREX1 would cause DNA damage. Here, we show that RVCL-associated TREX1 variants trigger DNA damage in humans, mice, and *Drosophila*, and that cells expressing RVCL mutant TREX1 are more vulnerable to DNA damage induced by chemotherapy and cytokines that up-regulate TREX1, leading to depletion of TREX1-high cells in RVCL mice. RVCL-associated TREX1 mutants inhibit homology-directed repair (HDR), causing DNA deletions and vulnerablility to PARP inhibitors. In women with RVCL, we observe early-onset breast cancer, similar to patients with BRCA1/2 variants. Our results provide a mechanistic basis linking aberrant TREX1 activity to the DNA damage theory of aging, premature senescence, and microvascular disease.

Age-related small vessel disease (SVD) can cause organ damage and disability, but the molecular mechanisms underlying SVD are incompletely understood. The pathophysiology of SVD involves damage to small arteries and capillaries, resulting in diminished blood flow to sensitive organs including the brain, eye, and kidney. This can ultimately lead to stroke, dementia, kidney failure, and blindness[1]. Various triggers of SVD include aging, hypertension, and genetic abnormalities[1–3]. Here, we report our studies of SVD-causing genetic variants in humans, mice, and *Drosophila*.

Retinal vasculopathy with cerebral leukoencephalopathy and systemic manifestations (RVCL or RVCL-S) is a dominantly inherited SVD that affects multiple organs, including the liver, kidney, retina, and brain[4]. Patients with RVCL develop symptoms affecting these organs between the ages of 35 and 55[5–8]. All patients with RVCL exhibit brain lesions and atrophy, and the large brain lesions often resemble tumors on magnetic resonance imaging (MRI)[5–8]. One half of family members are affected by the disease-causing mutations, leading to disability and premature death in 100% of affected individuals, often within 5–10 years of symptom onset[5–8].

RVCL is caused by heterozygous, dominant mutations in the three-prime repair exonuclease 1 (*TREX1*) gene, which encodes the most abundant mammalian 3′−5′ exonuclease in mammals[9]. In patients

✉e-mail: taisuke8077@bri.niigata-u.ac.jp; Jonathan.Miner@PennMedicine.UPenn.edu

with RVCL, a C-terminally truncated TREX1 exonuclease completely lacks the transmembrane domain (TMD), which anchors TREX1 to the endoplasmic reticulum (ER)[9]. Anchoring of TREX1 at the ER can prevent TREX1 from interacting with genomic DNA, and ER localization also allows TREX1 to degrade aberrant cytosolic DNA, thus negatively regulating cGAS-STING-type I interferon (IFN) signaling[10,11]. Indeed, TREX1 loss-of-function or dominant negative variants cause interferonopathy in humans and mice, including in patients with Aicardi-Goutières syndrome (AGS)[12–14]. Whereas some have proposed that RVCL mutant TREX1 may also cause interferonopathy[15,16], larger studies do not suggest systemic inflammation in RVCL[17,18]. Nevertheless, MRI imaging frequently reveals contrast enhancement of brain lesions, suggesting that local inflammation might contribute to disease. Thus, the role of local inflammation in RVCL pathogenesis remains to be fully defined.

Under certain conditions, full-length TREX1 can act on nuclear, genomic DNA[19–21]. Although TREX1 is normally excluded from the nucleus[19,21], wild-type (WT) TREX1 can also translocate to the nucleus to interact with nuclear enzymes including PARP1[22]. This suggests that TREX1 may participate in DNA repair in a highly regulated manner. For example, DNA-damaging agents cause full-length TREX1 to translocate to the nucleus[23], and TREX1 can remove 3′-DNA–peptide/protein crosslinks arising from abasic site adducts generated during DNA damage events[24]. This highlights a regulated role of WT TREX1 in DNA repair, including regulated translocation of TREX1 into the nucleus in specific circumstances[24].

Given the fact that TMD-deficient TREX1 is a fully functional but mislocalized exonuclease, we reasoned that RVCL-causing TREX1 variants may exhibit aberrant nuclear activity leading to chronic DNA damage, ultimately resulting in premature senescence or cell death. Furthermore, RVCL patients are usually healthy until the 4th or 5th decades of life, and this may imply that age-related cellular damage underlies the disease. A role for DNA damage in RVCL is also suggested by the fact that brain pathology of RVCL resembles radiation necrosis, a type of pathology associated with DNA damage induced by ionizing radiation[7].

Here, we demonstrate that TMD-deficient RVCL mutant TREX1 suppress homology-directed repair (HDR), leading to accumulation of DNA double-strand breaks (DSBs) in *Drosophila*, mice, and humans. We show that RVCL mutant TREX1 causes premature senescence in some cell types, and premature cell death in others. The DNA damage repair phenotypes of cells expressing TMD-deficient TREX1 resemble those observed in breast cancer patients with inherited *BRCA1/2* variants[25,26]. Consistent with our molecular findings, we demonstrate that cells expressing RVCL mutant TREX1 are more vulnerable to ionizing radiation and chemotherapy, which trigger a cycle of DNA damage, inflammation, cytokine-mediated TREX1 induction. Finally, we show that RVCL mutant TREX1 inhibits homology-directed repair (HDR) and causes DNA deletions upon induction of DNA double-strand breaks, and we demonstrate that women with RVCL have higher odds of developing breast cancer before age 45, likely because of underlying DNA damage repair defects. Our results have major implications for the care of patients with RVCL, and our discoveries provide insights into the mechanistic basis of RVCL and the role of TREX1 in small vessel disease and aging.

## Results

To better understand the molecular mechanisms of RVCL pathogenesis, we utilized the *Drosophila melanogaster* model, a well-established fruitfly system for genetic studies, including RNA interference (RNAi) screens[27]. We generated transgenic *Drosophila* lines expressing human wild-type (WT) TREX1, C-terminally truncated TREX1 (V235Gfs; RVCL TREX1), or enzymatically inactive RVCL TREX1 (ΔExo RVCL TREX1) (Fig. 1 and Supplementary Fig. 1a)[28]. *Drosophila* expressing human TREX1 exhibited the "rough-eye" phenotype, a developmental defect

that was more severe in fruit flies expressing RVCL mutant TREX1 (Fig. 1a, b). Notably, exonuclease-inactivation (ΔExo RVCL TREX1) mitigated the severity of the rough-eye phenotype induced by RVCL TREX1 (Fig. 1a, b). While WT TREX1 was predominantly localized in the cytoplasm, some WT TREX1 was found in the nucleus (Fig. 1c, d). By contrast, both RVCL and ΔExo RVCL TREX1 mutants were localized throughout the cell, including a much larger fraction in the nucleus (Fig. 1c, d), which suggests that nuclear localization of enzymatically active TREX1 exacerbates eye disease in *Drosophila*. The disease phenotype in *Drosophila* expressing WT TREX1 might be explained by over-expression, which might drive a certain amount of nuclear mislocalization of full-length TREX1. Although both WT and RVCL mutant TREX1 cause the rough eye phenotype in *Drosophila*, the disease is more severe in RVCL mutant flies (Fig. 1a, b), implying that higher levels of nuclear mislocalization of RVCL mutant TREX1 might be driving the rough eye phenotype. Indeed, the rough eye phenotype may be a consequence of DNA damage, since it also occurs upon knockdown of the DNA damage repair protein ATM[29].

To further elucidate the genetic pathways that influence the rough eye phenotype in the *Drosophila* model of RVCL, we conducted an RNAi screen of 367 genes encoding nuclear proteins (Supplementary Data 1). Knockdown of *Rad50*, a gene involved in regulating DNA damage repair pathway utilization[30], exhibited the second-highest rough eye suppression in flies expressing RVCL mutant TREX1 (Fig. 1e and Supplementary Data 1). Whereas *Rad50* deletion causes pupal lethality in *Drosophila*[31], *Rad50* knockdown is paradoxically protective of DNA damage phenotypes in *Drosophila*, including the rough eye phenotype in *ATM* knockdown *Drosophila*[29,32]. Indeed, reducing *Rad50* expression levels may prevent excessive activation of DNA damage repair pathways[29,32]. Our gene ontology (GO) pathway analysis on the 24 genes that ameliorated the rough eye phenotype of RVCL TREX1 flies revealed gene clusters in DNA damage response (DDR) pathways (Fig. 1f). Thus, RVCL-associated TREX1 mutants exacerbate the rough eye phenotype in *Drosophila*, perhaps by inducing DNA damage or dysregulating the DNA damage response.

To test whether RVCL TREX1 causes DNA damage in human cells, we established an inducible Flp-in expression system for TREX1 variants associated with several human diseases, including RVCL, systemic lupus erythematosus (SLE), Aicardi-Goutières syndrome (AGS), and familial chilblain lupus (FCL) (Fig. 2a and Supplementary Fig. 1b–c). We quantitated the formation of γH2AX and 53BP1 foci, markers of DNA DSBs (Fig. 2)[33]. All RVCL TREX1 mutants caused increased numbers of DSB foci; however, none of the TREX1 mutants associated with SLE, AGS, or FCL altered the frequency of DSBs (Fig. 2b, c)[12,14]. Large numbers of DSB foci required enzymatic activity and nuclear localization of RVCL TREX1 (Fig. 2b, c). Additionally, we detected higher levels of DNA-damage response (DDR) signaling, as indicated by the phosphorylation of ATM and Chk2 in cells expressing TREX1 mutations that cause RVCL, but not in cells expressing TREX1 variants associated with other diseases (Supplementary Fig. 2b, c). Again, we observed that enhanced DDR signaling required both the enzymatic activity and nuclear localization of RVCL TREX1 (Supplementary Fig. 2b, c), suggesting that aberrant nuclear activity of TREX1 causes DNA damage.

Since cell immortalization can alter the DDR, we created an inducible, stable TREX1 expression system in non-immortalized IMR-90 cells, a cell culture model of cellular senescence when cells are maintained within the Hayflick limit (Supplementary Fig. 3a)[34,35]. We found that IMR-90 cells expressing RVCL TREX1 had an increased number of DSB foci and increased DDR signaling (Fig. 2d–e and Supplementary Fig. 3b–e). Similar to the DNA damage observed in HEK293T cells, DNA damage induced by RVCL TREX1 in IMR-90 cells relied on nuclear localization and enzymatic function of TREX1 (Fig. 2d–e and Supplementary Fig. 3b–e). Furthermore, we found that RVCL TREX1 caused more DNA damage in the comet assay (Supplementary Fig. 3f–h). All of these results indicate that RVCL TREX1 causes

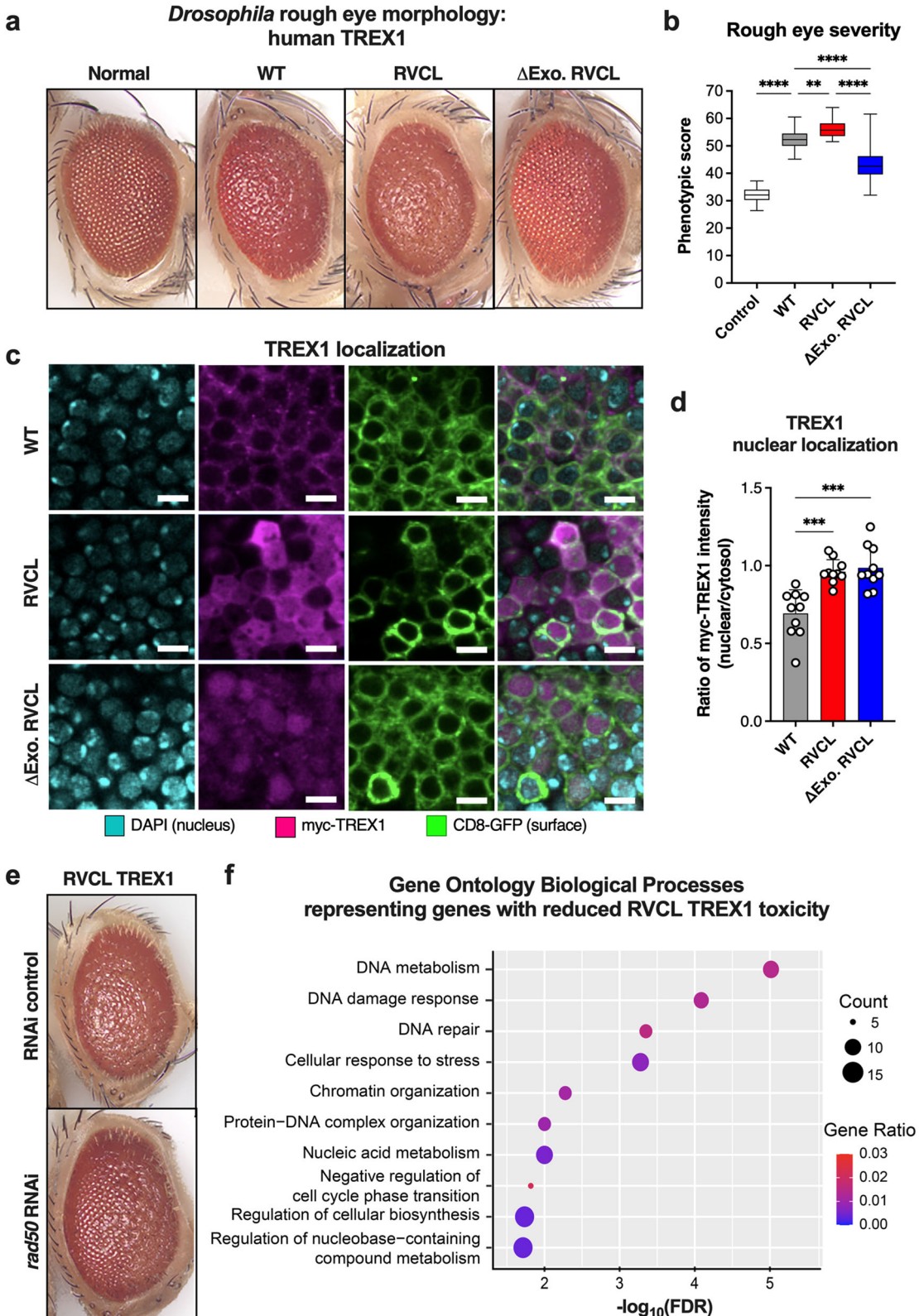

**a** *Drosophila* rough eye morphology: human TREX1

Normal | WT | RVCL | ΔExo. RVCL

**b** Rough eye severity

**c** TREX1 localization

DAPI (nucleus) | myc-TREX1 | CD8-GFP (surface)

**d** TREX1 nuclear localization

**e** RVCL TREX1

RNAi control | *rad50* RNAi

**f** Gene Ontology Biological Processes representing genes with reduced RVCL TREX1 toxicity

DNA damage. Furthermore, IMR-90 cells expressing RVCL TREX1 but not ΔExo RVCL or NES RVCL arrested proliferation and up-regulated genes related to the senescence-associated secretory phenotype (SASP) (Fig. 2f−g), another result consistent with a role for RVCL TREX1 in promoting DNA damage. Notably, WT TREX1 can respond to or cause DNA damage under certain conditions, including translocation to the nucleus with the SET complex[19,21]. Interestingly, the SET complex

was not detected in the nuclei of cells expressing RVCL TREX1 (Fig. 2h), suggesting a distinct mechanism of action for RVCL TREX1-induced DNA damage. Together, these findings suggest a mechanism by which the aberrant nuclear activity of TREX1 induces genotoxicity.

To understand the effects of C-terminally truncated TREX1 in mice, we generated a mouse model of RVCL that is heterozygous for the TREX1 T235Gfs mutation at the endogenous locus (Supplementary

**Fig. 1 | Disease-causing, C-terminally truncated TREX1 exacerbates the rough eye phenotype in *Drosophila*. a** Representative images of compound eyes from *Drosophila* over-expressing WT human TREX1 (WT), the TREX1 C-terminal frameshift mutant (RVCL), the TREX1 C-terminal frameshift mutant lacking exonuclease activity (ΔExo RVCL), and control (Normal). **b** Quantification of rough eye phenotypic scores from **a**. (*P* < 0.0001 for Control vs. WT, WT vs. ΔExo RVCL, and RVCL vs. ΔExo RVCL, *P* = 0.0033 for WT vs. RVCL) **c** Representative fluorescence microscopy in *Drosophila* Kenyon cells over-expressing WT TREX1, RVCL TREX1 (V235Gfs), or ΔExo RVCL with immunostaining for Myc (TREX1), DAPI (nucleus), and CD8-GFP (structural integrity). Scale bar = 4 μm. **d** Quantification of the ratio of nucleic to cytoplasmic TREX1 signal from **c**. (*P* = 0.0004 for WT vs. RVCL, *P* = 0.0005 for WT vs. ΔExo RVCL). **e** Representative image of compound eye of *Drosophila* over-expressing RVCL TREX1 with either control RNAi or the rough eye-modifying *radSO*

RNAi. **f** Over-represented GO terms enriched by genes with reduced toxicity of RVCL mutant TREX1 with respect to biological process classification. The plots are size-scaled by the number of effective genes enriched for each GO term and color-scaled by the gene ratio (ratio of the number of effective genes to the number of genes associated with the GO term). Data in **a**, **c**, and **e** are representative of independent experiments. Data in **b** represent *n* = 25 control, *n* = 36 WT, *n* = 29 RVCL, and *n* = 34 ΔExo RVCL. Data in **d** represent *n* = 10 for all groups. Boxes in **b** represent the 25th and 75th percentiles, with a solid line within the box showing the median value. Whiskers show the largest and smallest observed value. Data in **d** represent the mean ± SD. Data in **b** and **d** were analyzed one-way ANOVA with a two-sided Bonferroni post hoc test. Source data are provided as a Source Data file. **P < 0.01; ***P < 0.001; ****P < 0.0001.

Fig. 4a). Like patients with RVCL, these mutant mice produce both full-length and C-terminal-truncated TREX1 proteins (Supplementary Fig. 4b). We hypothesized that tracking TREX1 expression levels would be critical for studying the toxic effects of the RVCL TREX1 mutant, so we defined expression levels TREX1 protein in MEFs that express both WT and mutant TREX1 from the endogenous locus. Cells with defects in DDR have increased sensitivity to DNA-damaging agents[25], so we tested our hypothesis using a cell viability screen in non-clonal populations of WT and RVCL mutant MEFs treated with DNA-damaging agents such as PARP inhibitors. Treatment with the PARP inhibitor olaparib resulted in a 15–20% cell death in heterozygous RVCL MEFs compared to 3–6% death in WT MEFs (Fig. 3a, b). Olaparib also caused increased DNA damage in RVCL mutant MEFs, as indicated by γH2AX staining (Fig. 3c). However, unlike our flow cytometric data, the PARP inhibitor dose responses revealed more subtle effects of PARP inhibitors in non-clonal MEFs, despite the increased presence of dead cells in the culture (Fig. 3a, b and Supplementary Fig. 5a). However, we noticed that TREX1 expression is highly heterogeneous in cloned MEFs (Supplementary Fig. 5b). Therefore, we reasoned that vulnerability to PARP inhibitors depends on expression levels of the RVCL mutant TREX1. Indeed, in cultures of MEFs with heterogeneous TREX1 expression, we found that PARP1 deletion selected for MEFs with low expression of the TREX1 mutant (Supplementary Fig. 5c). Thus, RVCL TREX1-high cells are more vulnerable to PARP1 deletion.

Given the importance of TREX1 expression levels in our studies, we realized that we must control for TREX1 expression levels in studies of DNA damage. Therefore, we utilized mice expressing HA-tagged full-length or RVCL mutant, since the tag allows antibody staining to facilitate comparisons between TREX1-high and TREX1-low cells (Fig. 3d and Supplementary Fig. 6a, b)[36]. Treatment of primary mouse BMDMs with olaparib led to more up-regulation of γH2AX in RVCL TREX1-high cells compared to RVCL TREX1-low or WT TREX1-expressing BMDMs (Fig. 3e, f). DNA damage was blocked by a TREX1 inhibitor in RVCL mutant cells (Fig. 3g, h), suggesting that TREX1 enzyme activity is required for DNA damage. Next, we utilized the SensiTive Recognition of Individual DNA Ends (STRIDE) assay[37] to quantitate the number of DSBs in olaparib-treated BMDMs, thereby confirming higher numbers of DNA breaks in RVCL BMDMs compared with WT BMDMs (Fig. 3i, j). Thus, RVCL TREX1 renders primary BMDMs more vulnerable to a PARP inhibitor, further suggesting a role for RVCL TREX1 in promoting DNA damage. To test whether RVCL animals also exhibit greater sensitivity to PARP inhibitors, we injected RVCL mice and WT littermates with olaparib daily for two weeks. Histopathological examination revealed perivascular inflammatory lesions in the livers of RVCL mice but not WT littermates (Fig. 3k).

Olaparib is a chemotherapeutic agent[38], and another chemotherapeutic agent called aclarubicin was previously considered as a potential therapy for RVCL, including a Phase I clinical trial[39–41]. In our own experiments testing aclarubicin in MEFs, we observed that a subset of WT MEFs was resistant to aclarubicin (Supplementary Fig. 7a). In contrast, all RVCL MEFs died after treatment with

aclarubicin (Supplementary Fig. 7a). This further supports the idea that RVCL cells are more vulnerable to chemotherapy. In light of these observations, we reviewed patient case records, including those of patients previously treated with aclarubicin. We observed that treatment with aclarubicin did not halt disease progression. In fact, dose reduction was required because of weight loss and morbidity in multiple patients. Furthermore, based on MRI imaging of the brain taken as part of routine standard of care, disease progression was observed in the months following dose escalation of aclarubicin (Supplementary Fig. 7b-c). Collectively, these results are consistent with the idea that RVCL mutations create heightened sensitivity to the DNA-damaging effects of chemotherapy, and that chemotherapy might even worsen or accelerate disease.

Since high expression of the RVCL mutant TREX1 induces cytotoxicity, we next wondered what types of stimuli might regulate TREX1 expression. The promoters of both mouse and human *TREX1* have multiple putative binding sites for cytokine-responsive transcription factors, including STAT-binding sites, interferon (IFN)-stimulated response elements (ISRE), and NF-κB-binding sites (Fig. 4a). Therefore, we reasoned that interferons and pro-inflammatory cytokines would up-regulate TREX1. To begin to test this hypothesis, we treated cultured BMDMs with a type I IFN (IFN-β), type II IFN (IFN-γ), or lipopolysaccharide (LPS), and this led to significant up-regulation of TREX1 in response to all three stimuli (Fig. 4b, c). Next, we performed intraperitoneal injection of LPS to induce pro-inflammatory cytokines in mice, and this also caused rapid up-regulation of TREX1 in mice expressing TREX1 under control of the endogenous promoter (Fig. 4d). Thus, TREX1 is up-regulated in response to inflammatory signals including cytokines in mice.

Since the mouse and human *TREX1* promoters are both responsive to cytokines[42] (Fig. 4a), we tested cytokine levels in the sera of a large cohort of RVCL patients and age-matched healthy controls. Analysis of serum samples did not reveal any signs of systemic inflammation in patients with RVCL (Supplementary Data 2). Similarly, no statistically significant differences were found in interferon-stimulated gene (ISG) expression between the peripheral blood mononuclear cells (PBMCs) of RVCL patients and healthy controls (Supplementary Fig. 8).

Because RVCL is a late-onset disease, and since aging itself is associated with low-grade inflammation[43,44], we tested whether age-related TREX1 up-regulation coincides with age-related inflammatory gene expression in humans. We analyzed data from the publicly available voyAGEr database, which contains age-related transcriptomic data from 48 human tissues[45], and we found that *TREX1* expression increases with age in the brain and liver, which are two organs severely affected in patients with RVCL (Supplementary Fig. 9a). Indeed, we found that inflammatory genes increase with age in multiple organs including the arteries, the brain, and kidneys (Supplementary Fig. 9b–f), a result consistent with the theory that aging is associated with chronic, low-level inflammation[43,44], and implying a mechanism for age-related up-regulation of TREX1. We also confirmed that TREX1

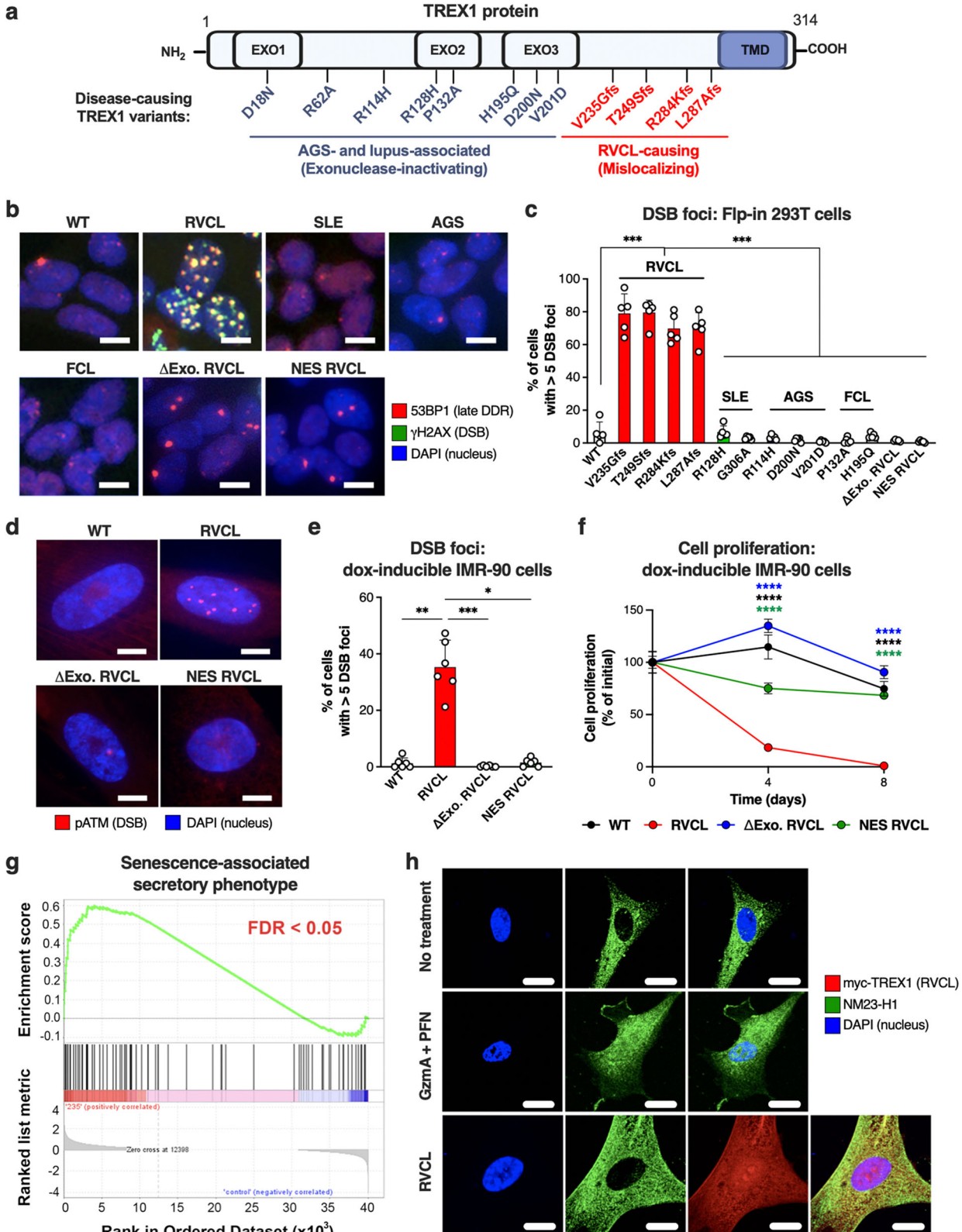

protein is up-regulated in the liver and brain of older adult mice compared to 1-month-old animals (Fig. 4e, f). Age-related up-regulation of TREX1 might explain why patients with RVCL become sick only around the 4th or 5th decades of life.

DNA damage causes inflammation, including the type I IFN response and production of pro-inflammatory cytokines[46,47], so we tested whether DNA damage might also cause up-regulation of TREX1,

and also whether TREX1-mediated DNA damage might cause low-level inflammation. Indeed, exposure to ionizing radiation caused more up-regulation of IFN-stimulated genes (ISGs) in RVCL MEFs (Fig. 4g), a result that may reflect cell-intrinsic immune activation due to DNA damage. Next, we confirmed that the topoisomerase inhibitor camptothecin (CPT) also causes more DNA damage in RVCL mutant BMDMs (Supplementary Fig. 10a-b), demonstrating that RVCL mutant BMDMs

**Fig. 2 | Aberrant nuclear activity of C-terminally truncated TREX1 triggers DNA damage and senescence in mammalian cells. a** Schematic depicting the distinction between locations of interferonopathy-associated *TREX1* variants and RVCL-causing *TREX1* variants. The exonuclease (EXO) and transmembrane (TMD) domains are indicated. **b** Representative confocal immunofluorescence images of Flp-in 293 T cells stably expressing WT TREX1 or disease-causing TREX1 mutants associated with RVCL, SLE, AGS, or FCL or RVCL TREX1 (V235Gfs) with a nuclear export signal (NES RVCL) or lacking enzymatic activity (ΔExo RVCL) with immunostaining for 53BP1 (red) and γH2AX (green). Scale bar = 15 μm. **c** Quantitation of cells with >5 γH2AX/53BP1 foci in cells from **b**. (*P* = 0.008) **d** Representative images of pATM foci in IMR-90 cells with doxycycline-inducible WT TREX1, RVCL TREX1, ΔExo RVCL, or NES RVCL. Scale bar = 10 μm. **e** Quantitation of frequency of cells with >5 pATM foci in cells from **d**. (*P* = 0.01 for WT vs. RVCL, *P* = 0.0007 for RVCL vs. ΔExo RVCL, *P* = 0.045 RVCL vs. NES RVCL). **f** Line graph of the change in proliferation rate over time of IMR-90 cells expressing WT TREX1, RVCL TREX1, NES RVCL, or ΔExo RVCL. The graph shows the data as 100% of the average value on day 0. (*P* = 0.0001). **g** Gene set enrichment analysis (GSEA) profile of SASP gene sets from RNA-seq performed in IMR-90 cells stably expressing RVCL mutant TREX1 (TREX1 V235Gfs) by doxycycline for 2 weeks and in cells without doxycycline. Analysis was performed with *n* = 4 replicates. FDR is highlighted in red. **h** Representative fluorescence microscopy images of IMR-90 cells with immunostaining for NM23-H1, DAPI (nucleus), and myc (TREX1). Data in **h** are representative of 3 independent experiments. Scale bar = 15 μm. Data in **b**, **c**, and **f** are representative of at least 2 independent experiments performed with *n* = 5 replicates per cell line. Data in **d**–**e** are representative of at least 2 independent experiments performed with *n* = 6 replicates per cell line. Data in **c**, **e**, and **f** represent the mean ± SD. Data in **c** and **e** were analyzed by one-way ANOVA with a two-sided Bonferroni post hoc comparison. Data in **f** were analyzed by two-way ANOVA. Source data are provided as a Source Data file. * *P* < 0.05; **P* < 0.01; ***P* < 0.001; ****P* ˂ 0.0001.

are more vulnerable to a variety of chemotherapeutic agents. After treatment of BMDMs with CPT and a blocking antibody against the type I IFN receptor (IFNAR1), we found that CPT causes IFNAR1-dependent up-regulation of both WT and RVCL mutant TREX1 (Supplementary Fig. 10c), confirming a role for IFNs in up-regulating TREX1 in response to DNA damage. Treatment of BMDMs with type I or type II IFN led to more DNA damage in RVCL BMDMs that express TREX1 from the endogenous promoter (Fig. 4h–j). This suggests a model in which RVCL mutant TREX1 may trigger a malignant cycle mediated by DNA damage, leading to inflammation, cytokine-induced TREX1 up-regulation, and further DNA damage (Fig. 4k).

Since we had discovered that TREX1 expression must be tracked with defining RVCL molecular phenotypes, we wanted to characterize endogenously expressed TREX1 levels in different cell populations in mice. TREX1 is a highly potent enzyme with very low expression levels in most cell types, making it very difficult to observe the heterogeneity of TREX1 expression among different cells. As a consequence, the cell type-specific expression patterns of TREX1 had not been clearly defined. Therefore, we studied TREX1-dsRed reporter mice that express TREX1 as well as dsRed under control of the endogenous TREX1 promoter (Fig. 5a). We found that TREX1-dsRed is highly expressed in certain myeloid cell subsets, including Ly6C+ monocytes and CD11c+ myeloid cells (Fig. 5b–d). In contrast, Ly6G+ neutrophils do not express the TREX1 reporter (Fig. 5e), and B cells, CD4+ and CD8+ T cells, and NK cells express very low levels of the reporter (Fig. 5f–i). Thus, within the hematopoietic compartment, TREX1 is most highly expressed within specific myeloid cell subsets.

Since TREX1 is up-regulated in response to inflammatory signals including IFNs, we transduced mice with adeno-associated virus (AAV) containing either the type I IFN, IFN-α2, or LacZ (Fig. 6a), and we confirmed IFN-α2 up-regulates ISGs in the liver of WT animals, but not in animals lacking the receptor for type I IFN (*Ifnar1*-/- mice) (Fig. 6b, c). Seven days after transduction of mice with AAV-IFN-α2, we observed partial depletion of TREX1-expressing myeloid cell subsets in RVCL animals but not in WT littermate control mice (Fig. 6d–f). Unlike TREX1-expressing monocytes and macrophages, neutrophils do not express TREX1 (Fig. 5e), and neutrophil numbers remained similar in WT and RVCL animals after transduction with AAV-IFN-α2 (Fig. 6g). Thus, inflammation causes partial depletion of TREX1-expressing myeloid cells in RVCL animals.

Next, we hypothesized that RVCL TREX1 may cause DNA damage by disrupting a specific DNA repair pathway. DSB repair occurs through three primary pathways: homology-directed repair (HDR), non-homologous end-joining (NHEJ), and alternative NHEJ (Alt-NHEJ)[48,49]. We simultaneously measured HDR and NHEJ at endogenous gene loci using CRISPR/Cas9-mediated cleavage with a single-stranded donor template for single-stranded template repair (SSTR), a type of HDR (Fig. 7a)[50,51]. The SSTR template was phosphorothioate-modified to resist TREX1-mediated degradation[52]. We observed a significant decline in SSTR efficiency in cells expressing the RVCL TREX1 mutant (Fig. 7b). This suppression was mitigated when RVCL TREX1 was modified by either addition of a nuclear export signal or by introducing an inactivating mutation in the exonuclease domain (Fig. 7c). Conversely, we observed increased NHEJ efficiency in cells expressing the RVCL TREX1 variant, similar to that reported in HDR-deficient cells (Fig. 7d, e)[53]. The effect of RVCL mutant TREX1 on NHEJ was also dampened by introducing a nuclear export signal or by eliminating the catalytic activity of TREX1 (Fig. 7f). Next, we performed long-read sequencing DNA sequencing to directly test whether the RVCL mutant TREX1 causes deletions at the DSB site. We found that large deletions were most prevalent in cells expressing the RVCL mutant TREX1, but that WT TREX1 also induced large DNA deletions, albeit to a lesser degree (Fig. 7g, h). This confirms that the RVCL mutant TREX1 is more genotoxic than WT TREX1, although WT TREX1 can also induce large deletions when expressed at high levels. However, only RVCL mutant TREX1 caused a high frequency of small DNA deletions (Fig. 7i), providing further support to the model that the RVCL mutant directly damages genomic DNA, perhaps by degrading 3′ overhangs to inhibit HDR (Fig. 7j).

Since RVCL mutant TREX1 is localized in the nucleus, we hypothesized that TREX1 may directly bind to chromatin to promote DNA damage. In subcellular fractionation experiments with a DNase digestion step, we confirmed that RVCL mutant TREX1 is indeed associated with chromatin (Fig. 8a), confirming that RVCL mutant TREX1 mislocalized to the nucleus even when an expression is driven by the endogenous promoter. Next, we performed co-immunoprecipitation of human WT and RVCL mutant TREX1 followed by mass spectrometry to identify potential TREX1-interacting proteins. We found that both WT and the RVCL-causing TREX1 V235Gfs mutant interact with many nuclear and chromatin-bound proteins, including those involved in DNA damage response, transcription, and nucleosomes (Fig. 8b, c). For example, WT TREX1 was previously reported to bind PARP1, even in the absence of chromatin[22], and we found that both WT and RVCL mutant TREX1 interact with PARP1, as well as with many other proteins involved in DNA damage repair (Fig. 8c). The high degree of overlap may be explained by the fact that WT TREX1 can translocate to the nucleus and interact with nuclear proteins under certain conditions[19–21], with the key distinction being that RVCL mutant TREX1 is constitutively in the nucleus (Fig. 8a). These interactions between TREX1 and chromatin-associated proteins may be direct or indirect through chromatin bridging. Thus, while both WT and RVCL mutant TREX1 can interact with nuclear proteins, only mutant TREX1 is constitutively bound to chromatin.

HDR mediates chromosomal cross-over and segregation during meiosis[54,55]. Some families with RVCL recently began choosing in vitro fertilization with preimplantation genetic testing. During the course of routine clinical care and preimplantation genetic testing, we incidentally made some unexpected findings. We observed an

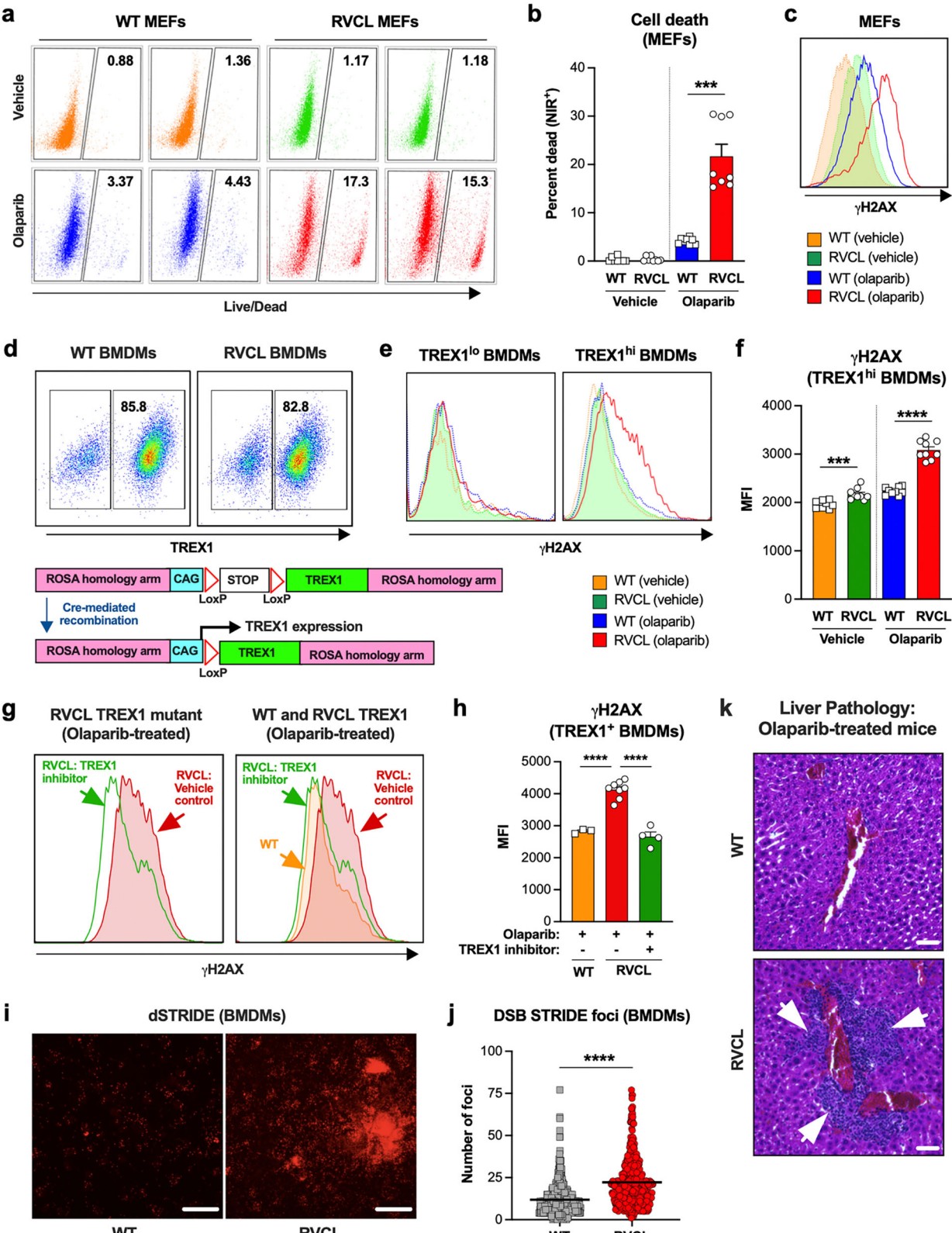

extraordinarily high rate of aneuploidy, including large chromosomal deletions, in four of seven embryos from a 27-year-old woman with RVCL (with the TREX1 D272fs mutation), at an age where such frequent aneuploidy is highly unusual (Fig. 9a)[56]. Interestingly, aneuploidy was observed in embryos regardless of whether they had inherited the pathogenic maternal *TREX1* variant. Early embryogenesis occurs prior to embryonic genome activation, but maternal proteins from the egg can persist in embryos that do not inherit

disease-causing mutations[57,58]. Both maternal and paternal chromosomes were involved, suggesting that some effects occurred during early cell divisions of embryogenesis. During routine SNP array analysis as part of the clinical standard of care, we also observed fewer HDR-dependent crossover events than expected based on datasets of embryos from non-RVCL patients (Fig. 9b, c)[56]. These observations are consistent with the model of TREX1-mediated inhibition of HDR.

**Fig. 3 | High expression of C-terminally truncated TREX1 causes vulnerability to DNA-damaging agents. a** Representative flow cytometric plots of live (NIR⁻) and dead (NIR⁺) WT and RVCL MEF cell populations following 72 h treatment with vehicle control (DMSO) or 10 μM olaparib. MEFs used in **a–c** express TREX1 out of the endogenous locus under control of the endogenous promoter. **b** Quantitation of percent dead (NIR⁺) cells from **a**. (*P* = 0.0002) **c** Representative histogram of γH2AX immunostaining in WT and RVCL (TREX1 T235Gfs) MEFs following 72 h of treatment with vehicle (DMSO) or olaparib (10 μM). **d** Representative flow cytometric analysis of TREX1ˡᵒ and TREX1ʰⁱ primary BMDM populations from these animals (upper panel) and a schematic of Cre-mediated recombination at the ROSA locus of Floxed-STOP WT TREX1 and V235Gfs TREX1 mice (lower panel). **e** Representative histograms of γH2AX expression of gated TREX1ˡᵒ and TREX1ʰⁱ WT and RVCL BMDMs treated with vehicle (DMSO) or olaparib (10 μM) for 72 h. **f** Quantitation of median fluorescence intensity (MFI) of γH2AX immunostaining of TREX1ʰⁱ WT and RVCL BMDM populations from **e**. (*P* = 0.0003 for vehicle, *P* < 0.0001 for olaparib). **g** Representative histograms of γH2AX in TREX1ʰⁱ WT and RVCL BMDMs treated with olaparib (10 μM) and TREX1 inhibitor (10 μM).

**h** Quantitation of MFI of γH2AX from **g**. (*P* < 0.0001) **i** Representative images of DNA DSB STRIDE analysis of WT or RVCL BMDMs treated with olaparib (10 μM) for 72 h. Scale bar = 10 μm. **j** Quantitation of nuclear DSB STRIDE foci in WT and RVCL BMDMs treated with 10 μM olaparib. (*P* < 0.0001). **k** Representative histological images of liver tissue sections stained with hematoxylin and eosin from WT litter mate control and heterozygous RVCL TREX1 (T235Gfs) mice treated with olaparib (40 mg/kg) for 14 days. Scale bar = 40 μm. Data in **a–c** and **k** are from mice expressing TREX1 from the endogenous locus under control of the endogenous promoter. Data in **d–j** are from mice expressing TREX1 under control of the CAG promoter. Data in **a, c, d, e, g**, and **k** are representative of at least 3 independent experiments. Data in **b, f**, and **h** represent the mean ± SEM of *n* = 8 samples per genotype from 2 independent experiments. Data in **i** are representative of 2 biologically independent experiments, each with multiple technical replicates. Data in **j** represent the mean ± SEM of WT *n* = 451, RVCL *n* = 494. Results in **b, f, h, j** were analyzed using two-sided Mann–Whitney test. Source data are provided as a Source Data file. ***P < 0.001; ****P < 0.0001.

Inherited mutations that impair HDR (e.g., *BRCA1* and *BRCA2* variants) are associated with elevated cancer risk, including early-onset breast cancer[59–61]. With this in mind, we reviewed clinical and family histories obtained during routine clinical care of patients with RVCL. Remarkably, three out of 19 female patients developed early-onset breast cancer, all diagnosed before the age of 45 years. The odds ratio (OR) for early-onset breast cancer in our RVCL patients is 8.74, compared to OR values of 18.8 and 9.3 for *BRCA1* and *BRCA2* variant carriers, respectively (Fig. 9d)[62]. Multiple other family members with unknown *TREX1* status had passed away from breast cancer at an early age, suggesting that this might be an underestimate of the true odds ratio. Thus, inherited C-terminal truncation variants in *TREX1* might cause increased odds of breast cancer, similar to the common risk alleles in *BRCA1* and *BRCA2*.

Collectively, our discoveries in *Drosophila*, mice, and humans suggest that TMD-deficient TREX1 variants cause genomic instability and disruption of DNA damage repair, and that this occurs via effects on HDR. These findings reveal a TREX1-mediated mechanism of DNA damage in human disease (Fig. 9e) and would explain why patients with RVCL develop small vessel disease that mimics the effects of ionizing radiation.

## Discussion

We found that TMD-deficient TREX1, which causes an inherited systemic small vessel disease called RVCL, inhibits HDR and promotes DNA double-strand breaks (DSB) and deletions. Inflammation and aging are associated with up-regulation of TREX1 in specific cell types, which may lead to accumulation of DSBs, cellular senescence, and loss of specific cell subsets. Thus, our results link TREX1-mediated DNA damage to age-related small vessel disease.

In our mouse model expressing TMD-deficient TREX1 under control of the endogenous promoter, we observed that DNA-damaging agents such as PARP inhibitors and cytokines led to an accumulation of DSBs and increased cytotoxicity. This vulnerability to PARP inhibitors mirrors the susceptibility observed in HDR-deficient cancer cells, such as BRCA1/2-deficient breast cancers[25,26]. Although we did not identify the precise molecular mechanism by which TREX1 promotes DNA damage, we confirmed that TMD-deficient TREX1 is constitutively localized in the nucleus and associates with chromatin, suggesting that the mutant TREX1 exonuclease may directly act on genomic DNA. DSBs in cells expressing TMD-deficient TREX1 were also induced upon treatment with other chemotherapeutic agents, as well as upon treatment with cytokines[63,64]. Collectively, our results suggest that RVCL-causing TREX1 mutants might not inherently produce DSBs, but rather disrupt the repair of newly formed DSBs in response to a variety of DNA-damaging agents. Using an established assay to quantify the efficiency of single-stranded template repair[50,65], a type of

HDR[50], we also demonstrated that TMD-deficient TREX1 decreased HDR, and that this effect of TREX1 depends on both nuclease activity and nuclear localization. Remarkably, we observed a reduced incidence of HDR-dependent genetic recombination events in embryos of a female patient with RVCL, as well as a high odds ratio of breast cancer in young adult women with RVCL. While these clinical observations are preliminary, the molecular basis for them is supported by our extensive cell culture and animal model results. Thus, TMD-deficient TREX1 is a potent suppressor of HDR and a mediator of DNA damage, leading to DNA damage phenotypes in multiple species.

There are multiple mechanisms by which TREX1 may cause DNA damage. Nuclear TREX1 may act at DNA breaks to degrade 3′ overhangs, which are critical for HDR as they facilitate strand invasion by aiding the entry of undamaged DNA strands[66,67]. In vitro experiments have previously demonstrated that TREX1 is capable of degrading 3′ single-strand overhangs[24,68], suggesting a mechanism by which TMD-deficient TREX1 may suppress HDR. In our systems, we found that RVCL mutant TREX1 causes both large and small deletions around the site of DNA double-strand breaks, consistent with the hypothesis of mutant TREX1 degrading 3′ overhangs. Alternatively, TREX1 may directly interact with or compete with DNA damage repair factors to inhibit HDR[22]. Indeed, we found that both WT and RVCL mutant TREX1 associates, either directly or indirectly, with chromatin-associated proteins including PARP1 and other DNA damage repair factors.

Surprisingly, we made the preliminary finding of an increased odds ratio of breast cancer in RVCL patients. Mutations in *BRCA1* and *BRCA2* are associated with an increased risk of breast cancer owing to the critical role of BRCA1/2 in DNA repair[62]. Mutations in these genes compromise HDR, leading to the accumulation of DNA damage and genomic instability, thereby increasing breast cancer susceptibility[69]. Thus, a high odds ratio of breast cancer in women with RVCL supports the hypothesis that TREX1-mediated DNA damage, likely occurring via disruption of HDR, increases the risk of breast cancer. We propose that carriers of RVCL-related *TREX1* variants should undergo earlier and more frequent screening due to the increased odds of developing breast cancer.

Our findings have significant therapeutic implications for patients with RVCL. Strategies to prevent DNA damage and inhibit TREX1 activity and TREX1 nuclear translocation are potential candidates for RVCL treatment. In this study, we demonstrated that a TREX1 inhibitor effectively reduces DNA damage[70]. The efficacy and potency of TREX1 inhibitors in preventing TREX1-mediated pathology in animals is under investigation and will be considered with caution. Indeed, genetic deletion of TREX1 also causes DNA damage via inflammatory pathways[71], suggesting that small molecular inhibition of the TREX1 enzyme may also carry risks of causing genomic instability via indirect mechanisms. In addition, caution is advised when exposing RVCL

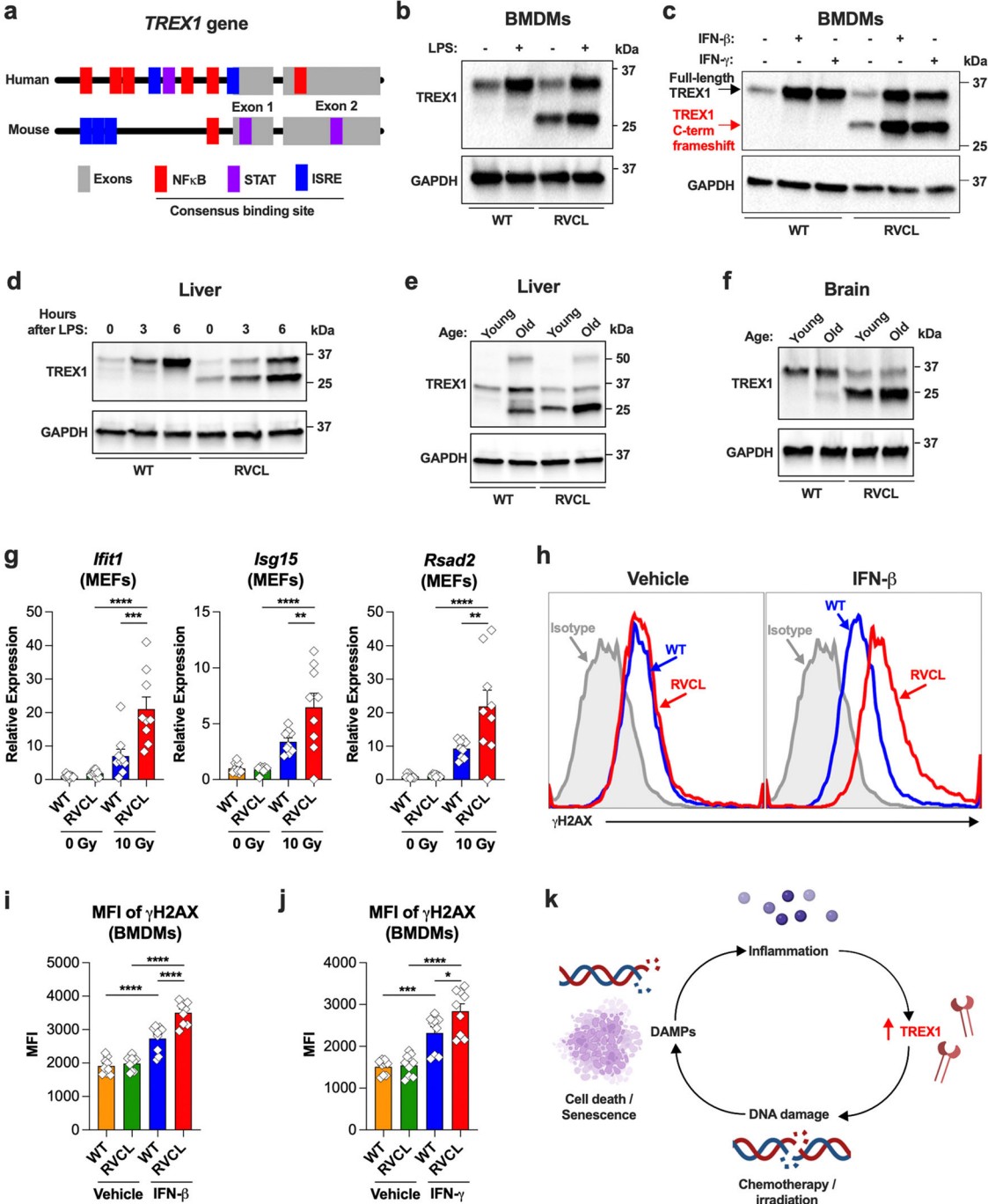

**Fig. 4 | Ionizing radiation and cytokines up-regulate TREX1 and cause excessive DNA damage in RVCL cells. a** Schematic of the *TREX1* gene showing the location of consensus sequences for NFκB binding sites, STAT binding sites, and Interferon-stimulated response elements (ISREs). **b** Representative Western blot of BMDMs from WT or heterozygous mice expressing RVCL mutant TREX1 under control of the endogenous promoter after 6 h of incubation with vehicle or LPS (1 μg/mL) with immunoblotting for TREX1 and GAPDH. **c** Western blots of BMDMS from WT or RVCL mice after 3 days of incubation with vehicle, IFN-β (100 IU/mL), or IFN-γ (10 ng/mL), followed by immunoblotting for TREX1 and GAPDH. **d** Western blot of liver from WT or heterozygous RVCL mice after intraperitoneal injection of vehicle or LPS (5 mg/kg), followed by immunoblotting for TREX1 and GAPDH. **e, f** Representative Western blot of liver (e) and brain (f) from WT or heterozygous RVCL mice at ages 1 month (Young) or 15 months (Old), followed by immunoblotting for TREX1 and GAPDH. **g** Quantitation of relative expression of the indicated ISGs by RT-qPCR in MEFs from WT or heterozygous RVCL mice 24 h after X-ray irradiation. ($P = 0.0001$ for RVCL 0 Gy vs. RVCL 10 Gy *Ifit1*, $P = 0.007$ for RVCL 0 Gy vs. RVCL 10 Gy *Isg15*, $P = 0.0031$ for RVCL 0 Gy vs. RVCL 10 Gy *Rsad2*,

$P < 0.0001$ for WT 10 Gy vs. RVCL 10 Gy). **h** Representative histograms of γH2AX expression of BMDMs from WT or heterozygous RVCL mice treated with vehicle or IFN-β (100 IU/mL) for 72 h. **i** Quantitation of mean fluorescence intensity (MFI) of γH2AX immunostaining from (h). ($P < 0.0001$) **j** Quantitation of MFI of γH2AX immunostaining of BMDMs form WT or heterozygous RVCL mice treated with vehicle or IFN-γ (10 ng/mL) for 72 h. ($P = 0.02$ for WT IFN-γ vs. RVCL IFN-γ, $P = 0.0002$ for WT vehicle vs. WT IFN-γ, $P < 0.0001$ for RVCL vehicle vs. RVCL IFN-γ). **k** Model of a malignant cycle of DNA damage and inflammation in cells expressing RVCL mutant TREX1. Created with BioRender.com released under a Creative Commons Attribution-NonCommercial-NoDerivs 4.0 International license. Data in **b–j** are from mice expressing TREX1 from the endogenous locus under control of the endogenous promoter. Data in **b–f** and **h** are representative of 2–3 independent experiments. Data in **g**, **i**, and **j** represent the mean ± SEM of $n = 9$ samples per genotype from 3 independent experiments. Data in **g**, **i**, and **j** were analyzed by one-way ANOVA with two-sided Šidák's multiple comparisons test. Source data are provided as a Source Data file. * $P < 0.05$; ** $P < 0.01$; ***$P < 0.001$; ****$P < 0.0001$.

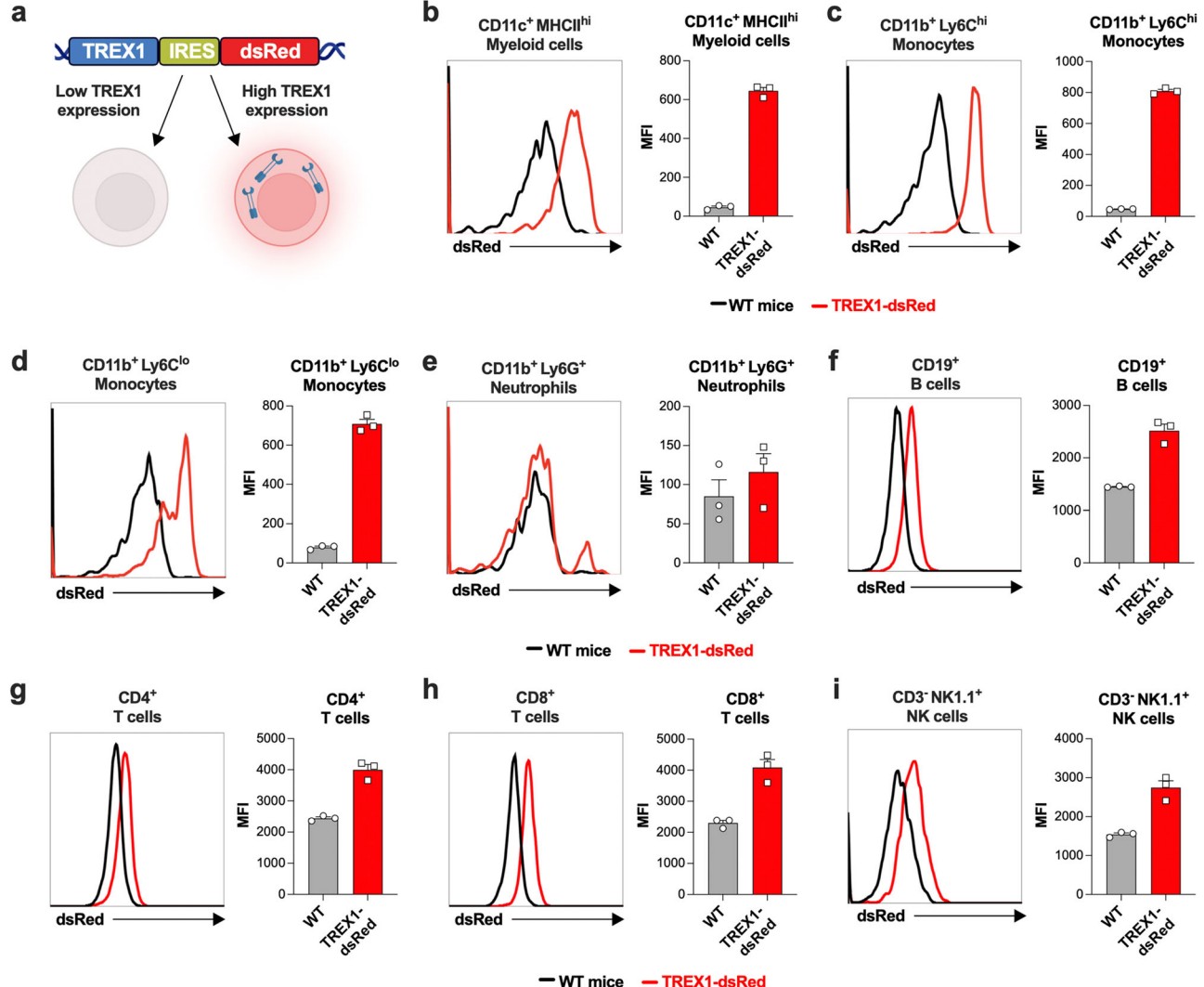

**Fig. 5 | CD11c⁺ myeloid cells and monocytes have high TREX1 expression in mice. a** Model of TREX1-dsRed reporter mice that express TREX1 and dsRed, separated by an IRES, at the endogenous TREX1 locus under control of the endogenous TREX1 promoter. Created with BioRender.com released under a Creative Commons Attribution-NonCommercial-NoDerivs 4.0 International license.

**b**–**i** Representative flow cytometric histogram of dsRed expression (left) with quantitation of mean fluorescence intensity (MFI) of dsRed (right) in the indicated cell types. Histograms are representative of $n = 3$ independent experiments. Error bars denote the SEM. Source data are provided as a Source Data file.

patients to genotoxic agents and radiation, as treatments that elevate TREX1 expression or induce DNA damage may exacerbate disease progression or trigger excessive inflammation. For instance, aclarubicin, a DNA-damaging agent, initially showed promise in reducing chemokine production in RVCL cells and quickly proceeded to a phase I clinical trial[39]. The notion that aclarubicin might be useful for RVCL was based on the idea that RVCL mutant TREX1 could trigger the production of aberrant sugars that cause inflammation, and aclarubicin had been reported to block the formation of these sugars[15]. We think sensitivity to aclarubicin likely reflects the vulnerability of RVCL cells to chemotherapeutic agents generally. In support of this, aclarubicin was toxic in patients with RVCL, dose reduction was necessary because of toxicity, no benefit was observed, and a phase II trial was not pursued. This fact, taken together with the heightened vulnerability of RVCL cells to aclarubicin, strongly indicates that chemotherapeutic agents are not a reasonable therapy for the disease. Furthermore, we have observed that many patients with RVCL have historically been treated with cyclophosphamide, another DNA-damaging chemotherapeutic agent that is commonly prescribed to patients with systemic autoimmune disease. Given that we found that

these patients may be particularly sensitive to DNA-damaging agents, we emphasize the need for a thorough risk-benefit evaluation before administering DNA-damaging agents to individuals harboring RVCL-related TREX1 mutations. Since we found that the genotoxicity correlates strongly with RVCL mutant TREX1 expression levels, more targeted therapies designed to diminish TREX1 expression deserve exploration as an approach to ameliorate this devastating disease.

Our results also indicate the potential risks of exposing patients with RVCL to ionizing radiation. The brain pathology in RVCL closely mirrors the features of radiation-induced necrosis, including vessel wall thickening, luminal narrowing, fibrinoid necrosis, adventitial fibrosis, and hyalinization in vessels[72]. Patients with DNA damage syndromes, including ataxia telangiectasia (AT), Nijmegen breakage syndrome (NBS), and Fanconi anemia (FA) develop brain lesions that can sometimes resemble those of patients with RVCL, and patients with AT, NBS, and FA have increased susceptibility to DNA damage from radiation[73–75]. RVCL patients might have a particularly high risk for radiation-induced tissue pathology since irradiation can up-regulate TREX1 expression and promote its nuclear translocation[22]. This raises the concern that irradiation could exacerbate the

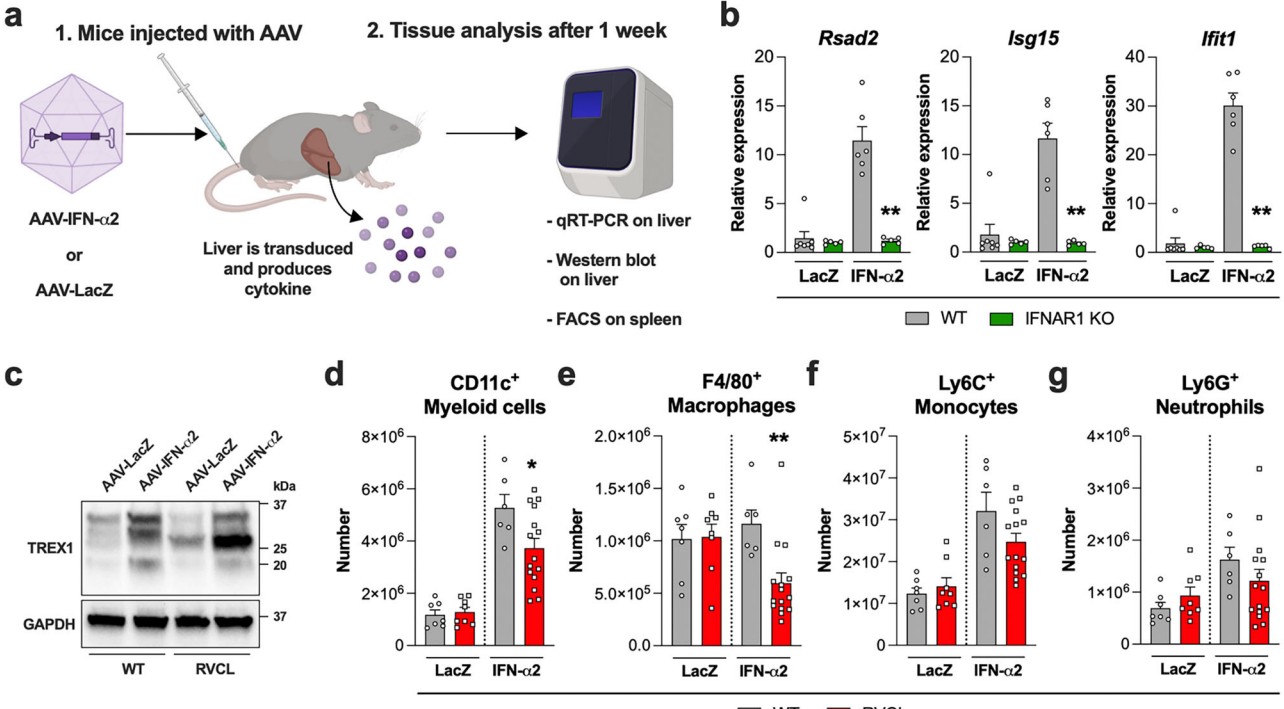

**Fig. 6 | IFN-α up-regulates TREX1 and causes loss of CD11c⁺ cells and macrophages in heterozygous RVCL mutant TREX1 mice. a** Model of experimental layout. Created with BioRender.com released under a Creative Commons Attribution-NonCommercial-NoDerivs 4.0 International license. **b** Quantitation of relative expression of the indicated ISGs in the liver of WT or IFNAR1 KO mice 1 week after injection with AAV-LacZ or AAV-IFN-α2. ($P = 0.0043$) **c** Representative Western blot of TREX1 and GAPDH from the livers of WT and heterozygous RVCL mice one month after injection with AAV-LacZ or AAV-IFN-α2. **d–g** Quantitation of cell number by flow cytometry from the spleen of WT or heterozygous RVCL mice one week after injection with AAV-LacZ or AAF-IFN-α2. ($P = 0.0351$ for d, $P = 0.0048$

for **e**). Data in **b–g** are from mice expressing TREX1 from the endogenous locus under control of the endogenous promoter. Data in **b** represent the mean ± SEM of $n = 7$ WT LacZ, $n = 5$ IFNAR1 KO LacZ, $n = 6$ WT IFN-α2, and $n = 5$ IFNAR1 KO IFN-α2 mice pooled from 2 independent experiments and were analyzed by Mann–Whitney $U$-test. Data in **c** are representative of 3 independent experiments. Data in **d–g** represent the mean ± SEM of $n = 7$ WT LacZ, $n = 8$ RVCL LacZ, $n = 6$ WT IFN-α2, and $n = 14$ RVCL IFN-α2 mice pooled from 2 independent experiments and were analyzed by two-sided unpaired $t$-test. Source data are provided as a Source Data file. *$P < 0.05$; **$P < 0.01$.

pathological state in RVCL patients by increasing the levels of TREX1, thereby amplifying DSBs. Indeed, we observed heightened inflammation upon irradiation of cells expressing RVCL mutant TREX1, likely reflecting inflammation in response to DNA damage. Alternatively, radiation-mediated ISG induction in RVCL cells may reflect increased inflammatory signaling, although this seems unlikely given the fact that C-terminal truncation of TREX1 does not activate the cGAS-STING pathway[17]. Therefore, extreme caution is advised when considering radiation procedures for patients with RVCL[6]. Even in individuals without RVCL-related mutations, radiation may result in elevated TREX1 levels and nuclear translocation, including nuclear translocation of full-length TREX1[23,76], leading to an increased incidence of radiation-induced DNA breaks. Thus, inhibition of TREX1 activity may be a potential strategy for mitigating radiation-induced brain injury including from recurrent CT scans.

In our study, we observed considerable variations in TREX1 expression levels when TREX1 is regulated by the endogenous promoter. We also found that higher TREX1 levels positively correlated with DNA damage. Furthermore, we found that the expression of TREX1 in the brain and liver, both of which are susceptible in RVCL, increased in an age-dependent manner[45]. These observations suggest that regulation of TREX1 expression by age and inflammation may contribute to organ-specific manifestations, late disease onset, or age-related progression in the brain. However, previous studies found that circulating mRNA levels of *TREX1* decrease with age in humans, although protein levels were not measured[77]. By contrast, we found that TREX1 protein levels increase with age in tissues. Thus, age-related regulation of TREX1 expression undoubtedly varies by cell and organ

type. Further investigation is warranted to elucidate the regulatory mechanisms governing TREX1 protein expression in different organs, as this may offer insights into these clinically relevant phenomena.

Mutations in TREX1 are associated with multiple different diseases including RVCL, SLE, FCL, and AGS[14]. Unlike what occurs in RVCL, dominant negative and loss-of-function mutations in TREX1 cause interferonopathies[12,13]. Although some studies have suggested that type I IFN signaling may be increased in RVCL cells under certain conditions[15,16], including the accumulation of single-stranded DNA in the cytosol and impaired glycosylation[15,16], these results might actually reflect byproducts of DNA damage. Indeed, single-stranded DNA is produced as a result of DNA break repair[78,79]. Thus, DNA damage may underlie the increased ssDNA and increased interferon signaling observed by others under certain conditions[16]. Nevertheless, we must emphasize that our findings, including serum cytokine analyses, do not support the idea that RVCL is a disease of systemic inflammation. Indeed, others have also demonstrated that patients with RVCL have a unique disease presentation without systemic elevation of type I IFN or ISGs[18]. Our results suggest an alternative possibility, which is that local inflammation triggered by DNA damage—rather than systemic hypercytokinemia—may play a role in this disease. However, the lack of systemic hypercytokinemia does not exclude a role of cytokines or local inflammation in RVCL. Indeed, patients with RVCL exhibit blood-brain barrier leakage of gadolinium contrast in MRI imaging, suggesting that inflammation occurs in some tissues during the course of disease. Whether inflammation is a consequence of DNA damage and injury, rather than the proximal cause of disease, remains to be determined. In RVCL, DNA damage might trigger activation of pattern

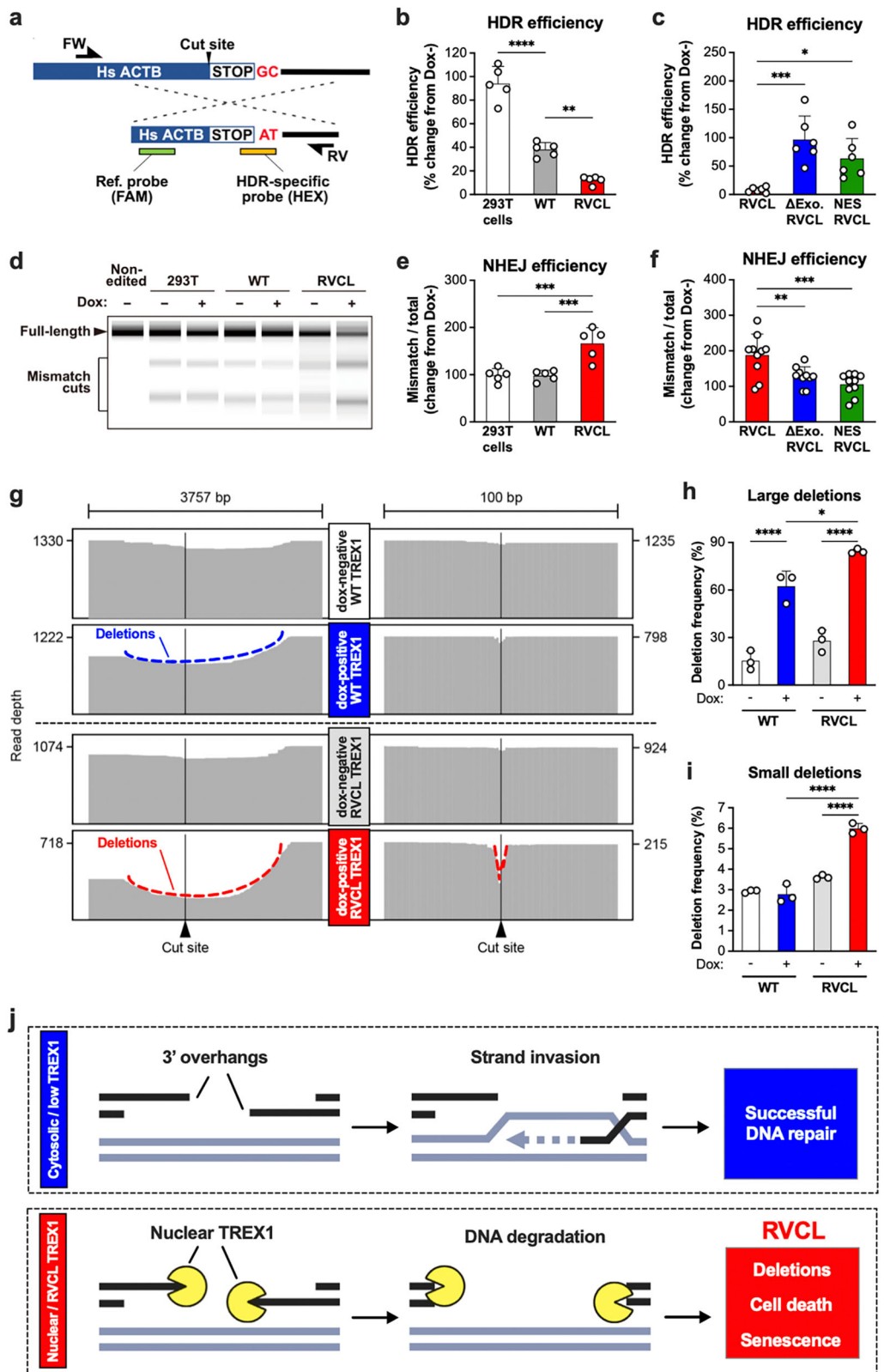

recognition receptors in specific cell types or tissues, leading to local cytokine-mediated up-regulation of TREX1, further DNA damage, and a malignant cycle of local inflammation.

　Although our work strongly suggests that RVCL is a DNA damage syndrome, many unanswered questions remain. First, our mouse models do not spontaneously reproduce systemic SVD, and there is still a need to create SVD models affecting the brain, kidney,

retina, and other organs. Another unanswered question involves the middle age-onset of RVCL, as well as the mechanisms underlying the progression of brain lesions[5–8]. One possibility for the delayed onset of the disease is that DNA damage accumulates over the lifespan of patients, causing local inflammation and depletion of important cell types, but only when TREX1 expression reaches a critical threshold. Physiological sources of inflammation including aging and infection

**Fig. 7 | Catalytically active, nuclear TREX1 disrupts HDR and enhances NHEJ.** **a** Schematic overview of the droplet digital PCR (ddPCR)-based HDR repair assay utilized in **b**, **c**. **b** Quantitation of HDR mediated repair via ddPCR assay of doxycycline-treated Flp-in 293 T cells expressing WT or RVCL TREX1. Data are expressed as percentage change in frequencies of HDR in doxycycline-treated cells relative to untreated cells. (*P* = 0.0017 for WT vs. RVCL, *P* < 0.0001 for WT vs 293 T cells) **c** Quantitation of HDR mediated repair via ddPCR assay of Flp-in 293 T cells expressing RVCL mutant TREX1, RVCL mutant TREX1 lacking enzymatic activity (ΔExo RVCL), or RVCL mutant TREX1 containing a nuclear export signal (NES REVCL) with data expressed as percentage change in frequencies of HDR in doxycycline-treated cells relative to untreated cells. (*P* = 0.0154 for RVCL vs. NES RVCL, *P* = 0.0004 for RVCL vs. ΔExo RVCL) **d** A representative ScreenTape electrophoretic gel image of PCR amplicons digested with mismatch cleavage nuclease from non-edited Flp-in 293T and CRISPR-Cas9-edited Flp-in WT TREX1, and TREX1 V235Gfs (RVCL) 293 T cells. **e** Quantification of NHEJ repair efficiency based on the mismatch cleavage band ratio in **d** with data expressed as percentage change in doxycycline treated cells relative to untreated cells. (*P* = 0.008) **f** Quantification of

NHEJ repair efficiency based on the mismatch cleavage band ratio of PCR amplicons digested with mismatch cleavage nuclease from RVCL, ΔExo RVCL, and NES RVCL Flp-in 293 T cells. Data are expressed as percentage change in doxycycline-treated cells relative to untreated cells. (*P* = 0.0058 for RVCL vs ΔExo RVCL, *P* = 0.0003 for RVCL vs NES RVCL) **g** Representative read alignments from long-read deep sequencing around the cut site from **a** in WT and RVCL Flp-in 293T cells before and after doxycycline induction. **h, i** Quantitation of the frequency of large (**h**) and small (**i**) deletions from **g**. (*P* = 0.0145 for WT Dox+ vs. RVCL Dox+ large deletions, *P* < 0.0001 for Dox- vs. Dox+ large deletions, *P* < 0.0001 for WT Dox+ vs. RVCL Dox+ and RVCL Dox- vs. RVCL Dox+ small deletions). **j** RVCL mutant TREX1 disrupts DNA damage repair by degrading 3' overhangs, leading to accumulation of deletions, cell death, and senescence. Data in **b**–**f**, **h**, and **i** represent the mean ± SD and are representative of independent biological replicates; *n* = 5 (**b**, **e**), *n* = 6 (**c**), *n* = 10 (**f**), or *n* = 3 (**h**, **i**). Data in **b**, **c**, **e**, and **f** were analyzed by ANOVA with Bonferroni post hoc comparison. Data in **h** and **i** were analyzed by ANOVA with two-sided Šidák's multiple comparisons test. Source data are provided as a Source Data file. *\*P* < 0.05; *\*\*P* < 0.01; *\*\*\*P* < 0.001; *\*\*\*\*P* < 0.0001.

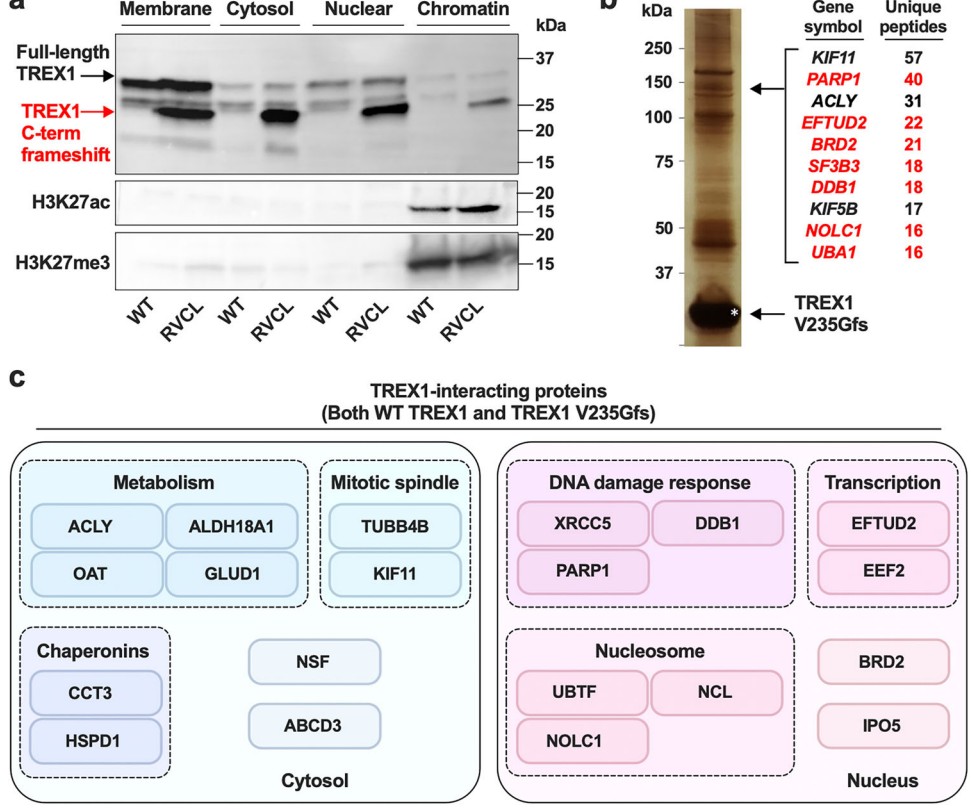

**Fig. 8 | RVCL mutant TREX1 interacts with chromatin and nuclear proteins involved in the DNA damage response.** **a** Representative Western blot after subcellular fractionation and SDS-PAGE of lysates from WT and heterozygous RVCL BMDMs followed by immunoblotting of TREX1, H3K27ac, and H3K27me3 from the indicated subcellular fractions. **b** A silver-stain of co-immunoprecipitated proteins using 3x-FLAG-tagged TREX1 V235Gfs as bait (left) and quantitation of unique co-precipitated peptides in the band that identified PARP1 and some of the other

interacting partners (right). Nuclear-localized proteins are indicated in red. **c** Summary diagram of WT TREX1- and TREX1 V235Gfs-interacting proteins with at least 20 unique peptides detected by immunoprecipitation-mass spectrometry. TREX1-interacting proteins are organized by their subcellular localization and cellular functions. Data in **a** are representative of 3 independent experiments. Data in **b** and **c** are from one mass spectrometry screen. Source data are provided as a Source Data file.

might cause TREX1-mediated DNA damage that accumulates over many years. In future studies, we will attempt to clarify the mechanisms underlying various types of small vessel lesions as well as large brain lesions, and why these brain lesions can sometimes progress rapidly in RVCL patients. In the *Drosophila* model and cultured cells, TMD-deficient TREX1 spontaneously caused DNA breaks. By contrast, cytokines or DNA-damaging agents were required to observe RVCL-associated DNA damage in primary cells, including in macrophages expressing TREX1 under control of the

endogenous promoter. We speculate that cell type-specific differences in TREX1 expression and DNA damage responses determine the various outcomes, including senescence or cell death. This underscores the need for further studies on the role of TREX1 in DNA damage repair in different cell types. Based on our finding that irradiation induces more inflammation in RVCL mutant cells, we speculate that RVCL patients also may be more sensitive to ionizing radiation, although additional data are necessary to confirm this hypothesis. Finally, we speculate that genomic DNA damage might

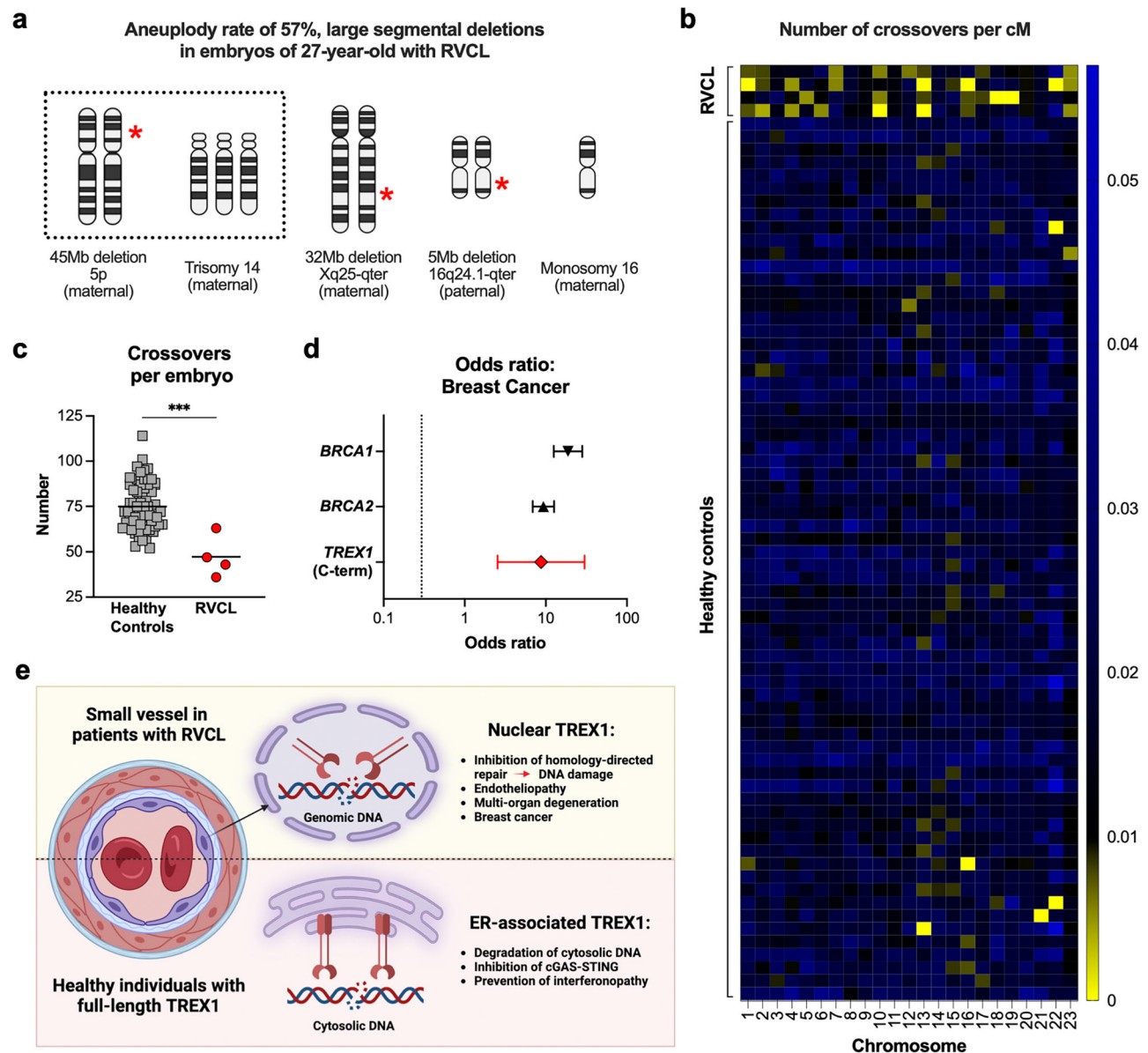

**Fig. 9 | Aneuploidy and reduced meiotic crossovers in embryos of a female RVCL patient and breast cancer in female patients with RVCL. a** Diagram representing the observed chromosomal aneuploidy in embryos of a 27-year-old RVCL patient undergoing in vitro fertilization with pre-implantation genetic testing. Red stars indicate location of deletion. **b** Heatmap showing the number of crossover events normalized to chromosome length (cM) per chromosome in healthy control and RVCL patient embryos. **c** A scatter plot showing the distribution of total crossover events in healthy control and RVCL patient embryos. ($P = 0.0004$) **d** Interval plot depicting the odds ratio (OR) of breast cancer in females with RVCL-causing *TREX1* variants compared with publicly available ORs of breast cancer in women with variants in *BRCA1* or *BRCA2*. **e** Model of RVCL disease pathogenesis. Created with BioRender.com released under a Creative Commons Attribution-NonCommercial-NoDerivs 4.0 International license. Data in **a**–**c** represent independent embryos analyzed as part of routine clinical care; healthy control embryo $n = 67$, RVCL embryo $n = 4$. Error bars in **d** represent 95% confidence intervals from $n = 19$ female patients with RVCL. Data in **c** analyzed by two-sided Mann–Whitney $U$ test. Source data are provided as a Source Data file. ***$P < 0.001$.

not be the sole mechanism of TREX1 toxicity in RVCL cells, although other mechanisms remain to be definitively proved.

In summary, our study demonstrates that RVCL causes DNA damage and disease phenotypes linked to genotoxicity, including heightened risk of breast cancer and sensitivity to DNA-damaging agents. These findings have substantial clinical implications for the management of patients with RVCL. Moreover, the observed accumulation of DSBs, attributed to the suppressive role of TREX1 in HDR, may have broader relevance, not only to RVCL but also to more general questions of inflamm-aging and post-irradiation injury to blood vessels. These discoveries underscore the opportunity to develop targeted strategies against TREX1 as therapies for both common and rare diseases.

## Methods
### Approvals
All protocols for animal studies were approved by the Institutional Animal Care and Use Committees (IACUC) or Institutional Review Boards (IRB) at the respective institutions. Serum samples of RVCL patients and healthy controls were IRB-exempt because the samples were anonymous and not linked to any clinical data. Other clinical findings were incidental to clinical care, including SNP results during preimplantation genetic testing, clinical histories of our patients, brain MRI assessments of patients with a history of prior treatment with aclarubicin, and cases of breast cancer discovered incidentally during the course of routine clinical care. Additionally, our patients gave

signed consent to participate in our longitudinal clinical studies including the IRB-approved REVEAL study at Penn.

## Fly strain

This study utilized several strains of flies, all of which were reared on standard fly food at a temperature of 25 °C. Among these strains was GMR-Gal4 (#1104), a Gal4 strain expressed in the fly eye during development. In addition, this study utilized a strain that possesses Gal4 and UAS sequences: c739-Gal4 and UAS-CD8GFP (#64305) expressed in the α/β lobes of Kenyon cells. As a negative control, the UAS-Luciferase RNAi strain (#31603) was used, which is an Inverted Repeat strain for the luciferase gene. UAS-dicer2 (#24648) was used to enhance RNAi efficiency. All strains were obtained from Bloomington Drosophila Stock Center (Bloomington, IN, USA). For the screening process, all genes categorized under "Chromosome" in the Flybase database were considered as candidates. A total of 367 RNAi strains were used in the screening, the details of which are listed in Supplementary Data 1. These strains were procured from the Vienna Drosophila Resource Center (Vienna, Austria).

## Establishment of transgenic Drosophila expressing human TREX1

For the wild-type human TREX1, as well as RVCL mutant TREX1 (TREX1 V235Gfs) and ΔExo RVCL mutant TREX1 (TREX1 R62A/V235Gfs), PCR was used to amplify the insert fragments. Using the DNA Ligation Kit Mighty Mix (Takara Bio, Shiga, Japan), these fragments were incorporated into the 20×UAS-IVS-P10 vector[80]. During this process, a Myc tag (EQKLISEEDL) was added to the N-terminus of each inserted human TREX1. The constructed recombinant vectors were then injected into fly embryos and inserted into the ZH86Fb landing site (Wellgenetics, Taipei, Taiwan).

## Evaluation of the rough eye phenotype

Each human TREX1 variant was expressed under the control of the GMR-Gal4 driver and the resulting flies were reared at 29 °C. After eclosion, adult flies were frozen and stored at -80 °C for later use. To photograph the compound eyes of the flies, a DP23 camera (Olympus, Japan) was attached to an Olympus BX53 microscope (Olympus, Japan) to capture images at 20x magnification. The area was photographed, including the compound eye, shifting the focus incrementally by 1.87 µm for each shot. These images were then depth-composited for each focus slice and the resulting phenotypic score was calculated using Flynotyper[81]. This phenotypic score reflects structural abnormalities in the compound eye, with a higher score indicating a greater degree of disorder in eye arrangement. A gene knockdown screen to alleviate the rough eye phenotype caused by the RVCL mutant TREX1 was performed using the following procedure. First, primary screening by visual inspection using at least three samples for each candidate gene was performed under blinded conditions. Based on the primary screening, 67 genes were identified. Furthermore, quantitative secondary screening was performed on these 67 genes using Flynotyper. Statistical analysis of the 67 samples was performed using Prism 9 software. For comparative analysis, one-way analysis of variance (ANOVA) was initially attempted. Subsequently, a nonparametric test with a Kruskal-Wallis test was performed to determine if there were statistically significant differences between the groups. GO enrichment analysis was performed on a group of genes with reduced toxicity of the mutant TREX1 (http://geneontology.org/). Assignment of human ortholog to fly genes was performed using the DRSC integrative ortholog prediction tool (DIOPT) (https://www.flyrnai.org/cgi-bin/DRSC_orthologs.pl).

## Immunohistochemistry and imaging of Drosophila

Immunohistochemistry was performed as previously described[82]. The following antibodies were used: mouse anti-myc (9B11) (1:2,000; Cell Signaling Technology, CST, 2276 S), rabbit anti-Histone H2AvD pS137 (Rockland, 600-401-914), anti-mouse Alexa Fluor 568 (Thermo Fisher Scientific, A-11004) and anti-rabbit Alexa Fluor 633 (Thermo Fisher Scientific, A-21071). The specimens were mounted using the Vectashield mounting medium (Vector Laboratories). To visualize the nucleus, 4′,6-diamidino-2-phenylindole (DAPI, Bio-Rad, 1351303) was used. Images were scanned using an FV3000 confocal microscope (Olympus, Tokyo, Japan) and captured using Imaris software (Oxford Instruments, Zurich, Switzerland). To quantify the localization pattern of each human TREX1 transgene, one- to two-day-old adult female flies were dissected. Referring to the DAPI signal, the region of the nucleus was selected using freehand selection in Fiji, an open-source image analysis software[83]. The nucleus was selected by experimenters who were blind to the genotype. The average fluorescence intensities of the nuclear and non-nuclear regions were calculated. To quantitatively evaluate the localization of myc-tagged human TREX1, the range stained by DAPI was considered the intranuclear signal and compared with the extranuclear signal to calculate the ratio of nuclear to cytoplasmic TREX1.

## Ribonucleoprotein electroporation for HDR induction

To examine the efficiency of HDR and NHEJ, the ribonucleoprotein (RNP) complex was electroporated into the Flp-In 293 T cells. Briefly, the gRNA complex was prepared by mixing equal amounts of Alt-R CRISPR Cas9 crRNA and tracrRNA at equal concentrations (50 µM) in duplex buffer (IDT), heated 95 °C for 5 min, and allowed to cool to room temperature. The prepared gRNA (120 pmol) was mixed with 100 pmol Alt-R Cas9 Nuclease V3 and allowed to form an RNP complex for 5 min at room temperature. Next, 4 µl of RNP complex was added to $1 \times 10^6$ cells suspended in 100 µl Opti-MEM with 1.2 µl of 100 µM (120 pmol) electroporation enhancer (IDT) and 1 µL of 100 µM (100 pmol) HDR single-strand donor oligo and incorporated by electroporation using a NEPA21 electroporator (NEPAGene). For single-strand donor oligos, ultramer DNA oligonucleotides were used in which the bonds between the five bases at the 3′ and 5′ ends were modified with phosphorothioate to protect from degradation by nucleases, including TREX1[52]. In addition, a two-base mutation immediately upstream of the stop codon was introduced into the donor oligo to detect the HDR repair allele (Fig. 7a) specifically. The detailed target gene recognition sequences of the crRNA and single-stranded donor oligo sequences are as follows: crRNA recognition sequence for the human ACTB gene (CCGCCTAGAAGCATTTGCGG)[84]. Donor oligo: (C*C*G*T*GTGGATCGG CGGCTCCATCCTGGCCTCGCTGTCCACCTTCCAGCAGATGTGGATCA GCAAGCAGGAGTATGACGAGTCCGGCCCCTCCATCGTCCACCGCAAA TGCTTCTAGATGGACTATGACTTAGTTGCGTTACACCCTTTCTTGACA AAACCTAACTTGCGCAGAAAACAAGATGAGATTGGCATGGC*T*T*T*A: phosphorothioate bonds are indicated by the asterisk (*) between the bases.). After electroporation, the cells were divided into two groups, one of which was induced to express TREX1 by the addition of doxycycline. Seventy-two hours after electroporation of RNP, genomic DNA was extracted and cleaned using a DNeasy Blood & Tissue Kit (Qiagen) and AMPure XP (Beckman coulter). The effect of each mutated TREX1 on HDR and NHEJ efficiency was assessed as the percentage change from dox-free cells between split pairs after electroporation.

## Quantification of HDR efficiency using droplet digital PCR

In this assay, the primer was set upstream of the complementation region with the donor oligo and inside the donor oligo, so that the mutation within the donor oligo was included in the amplified product. To quantify the total PCR amplification product and HDR allele-derived product in the same amplicon, two types of probes were designed in a single amplicon (Fig. 7a). The first is a FAM-labeled reference probe that does not overlap with the cut site and always binds to the amplicon regardless of the presence of the mutation insertion. The second is

a HEX-labeled probe that binds specifically to amplification products derived from alleles undergoing HDR repair. To ensure specificity of the donor oligo-derived mutant sequence by HDR repair, this probe is chimerically modified with locked nucleic acids (LNA). This HDR probe-positive amplicon was detected only when the donor oligo was introduced during genome editing. The detailed sequences of the primers and single-stranded donor oligo are as follows: FW primer (CTTCCCTCCTCAGATCATTGC), RV primer (AGAAAGGGTGTAACGC AACTAA), reference probe (CTGTCCACCTTCCAGCAGATGTGG), and HDR-specific probe (CTT + CTAG + A + T + GGA + C + TAT; LNA modifications are indicated by the plus sign(+) to the right of the modified base). Droplets enclosing the PCR mix were formed using the Bio-Rad QX-100 emulsification device. After PCR cycling, droplets were analyzed immediately using QuantaSoft v.1.6 (Bio-Rad). Frequencies of HDR in each sample are calculated as $F_{HDR} = C_{HDR} \times 100/C_{total}$, where $F$ = allelic frequency (%), $C_{HDR}$ = the HDR-repaired allelic (HDR-specific probe positive) concentration, and $C_{total}$ = the total allelic (reference probe positive) concentration (Fig. 7a).

## Mismatch cleavage assay

The genomic DNA sequence region containing the CRISPR Cas9 cut site was amplified by PCR using PrimeSTAR GXL DNA Polymerase. PCR products were denatured by heating at 95 °C for 5 min and then cooled using a thermocycler at rates of 2 °C/s to 85 °C and 1 °C/s to 25 °C to form heteroduplexes. Heteroduplex DNA digestion was performed using Guide-it Resolvase (TaKaRa) for 30 min at 37 °C. Digested PCR amplicons were separated by capillary gel electrophoresis and uncut amplicons and cleaved fragments were detected and quantified using an Agilent 2200 TapeStation system. The NHEJ efficiency was calculated as the ratio of the cleaved fragment signal to the total DNA fragment signal.

## Plasmid construction

The full-length coding region of human TREX1 cDNA was amplified by PCR from a human cDNA library (Clontech) with an N-terminal myc-His tag and subcloned into the multiple cloning site of the pcDNATM5/FRT/TO vector (BamHI/XhoI)(Thermo Fisher) or pRetroX TRE 3G vector (BamHI/MluI) (Clontech). Each mutant construct was produced using the GeneArt site-directed mutagenesis system (Thermo Fisher)[14,28,85]. A consensus MAPKK NES (NLVDLQKKLEELELDEQQ)-fused TREX1 was created by PCR using specific primers from the cDNA library and subcloned into the pcDNATM5/FRT/TO vector[86]. The primer sequences used by GeneArt to construct the mutant plasmid are indicated in Supplementary Data 3.

## Establishment of cells with stable gene expression

Stable gene-expressing cells were produced using the Flp-In system (Thermo Fisher) or Retro X Tet-On 3 G Inducible Expression System (Clontech). In the Flp-In system, T-Rex-293 Cells (Thermo Fisher) were seeded at approximately 60% confluent in 35 mm dishes one day before transfection. Then, 150 ng of pcDNA5/FRT/TO vector with each human TREX1 mutant cDNA and 850 ng of pOG44 was transfected into T-Rex-293 Cells by X-tremeGENE HP DNA Transfection Reagent (Roche). The medium was replaced with a selection medium containing 250 μg/mL hygromycin B. Each surviving colony was picked up and used for each experiment after confirming normal growth capacity and doxycycline-induced gene-of-interest expression.

The RetroX Tet-On 3 G inducible expression system was established in two steps. For retrovirus production, GP2-293 packaging cells were seeded at 60% confluence in 100 mm dishes 24 h before transfection. pAmpho vector (15 μg) and pRetroX Tet-3G vector (15 μg) or pRetroX TRE-3G vector (15 μg) with each human TREX1 mutant cDNA were transfected into the cells using the Xfect Transfection Reagent (Takara). After 4 h of incubation, the medium was replaced with a fresh medium. The medium was collected after 48 h and 72 h. After filtration

of the harvested medium to remove cell debris, a Retro-X concentrator was added to the medium and incubated overnight 4 °C. The processed medium was centrifuged at 1500×g for 45 min at 4 °C and the supernatant was discarded. The precipitated virus products were dissolved in a fresh medium. For retroviral transfection into target cells, $1.4 \times 10^5$ IMR-90 cells were seeded in 6-well plates 24 h before transfection. The seeded cells were co-transduced with RetroX-Tet3G and RetroX-TRE3G retroviruses at a ratio of 2:1 with polybrene (8 μg/ml) and centrifuged at 1500×g for 60 min at room temperature, followed by overnight incubation in a humidified incubator. After virus transduction, the cells were exposed to 2 μg/ml puromycin (Thermo Fisher) and 600 μg/ml geneticin (Thermo Fisher) for 6 days and used for each experiment. Cells were cultured in Dulbecco's modified Eagle's medium (DMEM, Gibco) supplemented with 10% Tet System Approved FBS (Clontech). As a positive control to analyze the behavior of the SET complex, IMR-90 cells were treated with granzyme A (2.5 μM) and perforin (40 ng/ml) for 20 min before analysis. All the cells were negative for mycoplasma.

## Neutral comet assay

Cellular DNA damage was assessed by a neutral comet assay using a comet assay kit (Trevigen). LMAgarose was melted by heating prior to the experiment. Briefly, 1 μl of SYBR Gold was dissolved in 30 mL of Tris-EDTA buffer (pH 7.5). The dissociated cells were adjusted to $1 \times 10^5$ per ml with cold PBS. 30 μl of the cell solution and 300 μl of Comet LMAgarose were combined and 50 μl of the mixture was evenly placed on the designated slides. The slides were placed on ice for 5 min and incubated at 4 °C for 30 min. Afterwards, the slides were incubated in lysis solution at 4 °C for 60 min and then in 1×Neutral Electrophoresis Buffer (50 mM Tris, 150 mM Sodium Acetate, pH 9.0) at 4 °C for 30 min without light exposure. Electrophoresis was conducted in 1×Neutral Electrophoresis Buffer at 4 °C and 17 volts for 45 min. Slides were incubated in DNA precipitation buffer (1 M ammonium acetate in ethanol) at room temperature for 30 min and then in 70% ethanol at room temperature for 30 min. The slides were dried and incubated with diluted SYBR Gold [1:30,000 in 10 mM Tris-HCl, 1 mM EDTA (pH 7.5)] for 30 min without light exposure. The slides were observed under an all-in-one microscope (Keyence; BioRevo BZ-9000) with a ×40 objective lens. These images were analyzed using OpenComet, an open-source plugin for the image-processing program ImageJ. 20–30 cells were analyzed per assay, and five independent assays were performed.

## EdU proliferation assay

Proliferation of IMR-90 cells was determined using the Click-iT EdU Imaging Kit (ThermoFisher) according to the manufacturer's protocol. Briefly, IMR-90 cells with inducible TREX1 expression were seeded at 50% confluency one day before adding EdU solution. Half of the medium was replaced with 10 μM EdU solution and the cells were incubated in a humidified incubator for 24 h. The cells were fixed with 4% paraformaldehyde (PFA) (Fujifilm) in phosphate buffer solution for 15 min and permeabilized with 0.5% Triton X-100 (Sigma Aldrich) for 20 min. EdU and azide were reacted with a copper catalyst for 30 min. Cell nuclei were visualized using Hoechst 33342 (Thermo Fisher). The percentage of EdU-positive cells relative to doxycycline-untreated cells, after doxycycline-induction of the indicated form of TREX1 in IMR90 cells, was calculated each day after doxycycline addition.

## Mice

A targeting sgRNA was designed with specificity to the location of the codon encoding threonine 235 of mouse TREX1, and site-specific cleavage was assessed in vitro using previously described methods[87]. spCAS9 and sgRNAs were in vitro-transcribed from PCR amplicons purified by Qiaquick PCR purification spin columns (Qiagen) using the MEGAshortscript T7 kit (Thermo Fisher) for sgRNA and mMESSAGE

mMACHINE T7 ultra kit (Thermo Fisher) for Cas9 mRNA. In vitro-transcribed Cas9 RNA was purified via lithium chloride precipitation and sgRNA was purified using the MEGAclear RNA purification kit (Life technologies) and diluted in nuclease free injection buffer. The TREX1 T235Gfs frameshift was introduced using a single-stranded DNA oligonucleotide donor (ssODN) which include the 5 aberrant C-terminal amino acid residues found in human TREX1 V235Gfs with 99 nucleotide flanking sequences homologous to the TREX1 open reading frame synthesized by IDT (ultramer oligo with the inserted nucleotides in uppercase): 5′(ctactgcagtgggtggacgaacatgcccggcccctttagcaccgtcaag cccatgtacggcGGTCACA-GCCTGTGTTAGTAAactccggctaccactggaacaac caacctaaggccacatgctgccacagctactacaccc)3′. Four-week-old female C57BL/6 J mice were super-ovulated with pregnant mare serum gonadotropin (PMSG) and human chorionic gonadotrophin (HCG) and mated with C57BL/6 J male mice. Fertilized single-cell embryos (embryonic day 0.5 [E0.5]) were isolated and microinjected with a combined mixture of 50 ng/µl Cas9, 25 ng/µl sgRNA and 100 ng/µl ssODN or 25 ng Cas9, 13 ng/µl sgRNA and 100 ng/µl ssODN in Dnase/Rnase free microinjection buffer (1 mM Tris, 0.25 mM EDTA, pH 7.4). Following microinjection -80-100 modified embryos per day, over a period of 6 days, were transferred into E0.5 pseudo-pregnant ICR/CD1 female recipient mice. Colonies were established from two independent founder mice backcrossed to wild-type C57BL/6 J mice for five generations, wild-type litter control mice were used for all experiments. Experiments were performed on mice of both sexes between (2 – 12 months) of age, matched with littermate control animals. Sample sizes were derived from previously published studies. Mice were randomly allocated for all experiments. Human TREX1 mutant mice were generated at Cyagen/Taconic as previously described[88]. Briefly, a CAG promoter, a transcriptional stop sequence flanked by loxP sequences, and N-terminal HA-tagged WT or RVCL human TREX1 cDNAs were cloned into ROSA26 targeting vectors. In vitro-transcribed Cas9 mRNA, sgRNAs, and targeting vectors were microinjected into fertilized embryos and implanted into C57BL/6 J mice. Colonies were established after backcrossing founder mice to wild-type C57BL/6J mice for five generations. Expression of TREX1 protein in macrophages was achieved by crossing the floxed-STOP TREX1 mice to transgenic LysM-Cre animals (Jax 004781). TREX1-dsRed reporter mice were produced at Ozgene and provided as a gift from the late Herbert C. Morse III (NIH). For olaparib injections, mice were injected intraperitoneally (IP) with 40 mg/kg of olaparib (Selleck Chemicals) in 30% polyethylene glycol (PEG) 300 (Selleck Chemicals) daily for 14 days. For AAV transduction, AAV8 encoding IFN-α2 or LacZ was produced at the Gene Therapy Program (University of Pennsylvania). Mice were intravenously injected with $1 \times 10^{11}$ genome copies of AAV in PBS and euthanized 1 week later for tissue analysis. Mice were housed in pathogen-free mouse facilities at the University of Pennsylvania Perelman School of Medicine and fed a standard diet and water *ad libitum*.

## Cell isolation

Spleens were kept on ice in Dulbecco's modified eagle medium (DMEM; Gibco, 11995081) supplemented with 10% fetal bovine serum (FBS), 2 mM L-GlutaMAX (Gibco, 35050061), 1X non-essential amino acids (Gibco, 11140050), 1 mM sodium pyruvate (Gibco, 11360070), 10 mM HEPES (Gibco, 15630080), 100 U/mL penicillin, 100 mg/mL streptomycin (Gibco, 15140122) (D10). To obtain single-cell suspensions, organs were mechanically disrupted and passed through 70-mM cell strainers and rinsed with 20 mL PBS. Red blood cells (RBCs) were lysed with the addition of 2 mL ACK lysis buffer (Gibco, A1049201) for 3 min, the cell suspensions were washed once with PBS (Gibco, 14190136) supplemented with 2% fetal bovine serum (Omega scientific, FB-01) (FACS buffer). To Isolate bone marrow cells for bone marrow derived macrophage differentiation, femurs and tibias were dissected from animals and the marrow flushed from bones. Debris were removed by filtration through a 70-µm cell strainers.

## Cell culture

To generate BMDMs, $5 \times 10^{6}$ marrow cells were cultured for 37 °C in 5% $CO_2$ in 10-cm Petri dishes with D10 with 40 ng/ml of macrophage colony-stimulating factor (M-CSF) (PeproTech, 315-02). On day 3 macrophages were fed with 5 mL complete DMEM containing 40 ng/mL M-CSF. Cell lines were cultured in D10 and incubated at 37 ˚C in 5% $CO_2$.

## SDS-PAGE and Western blotting

BMDMs, primary MEFs and HEK 293 T cells were solubilized in 1x RIPA buffer (CST, 9806) supplemented with protease and phosphatase inhibitors (Thermo Fisher, A32959). For tissue lysates, organs were homogenized in 1x RIPA buffer with protease and phosphatase inhibitors. The protein content of lysates was quantitated via BCA assay (Thermo Fisher, 23225) and equal amounts of protein were mixed with 2x Laemmli sample buffer (Biorad) with 5% β-mercaptoethanol (BME; Sigma Aldrich) and boiled at 95 °C for 5 min. Denatured samples were loaded on 4-20% SDS PAGE gradient gels (Biorad), then transferred to polyvinylidene fluoride membranes (EMD Millipore) via semi-dry transfer. Following transfer, membranes were blocked for 1 h at room temperature with 3% BSA or 5% non-fat milk in TBST. Resolved proteins were immunoblotted using primary antibodies against γH2AX (CST, 2577), H2AX (CST, 2595), GAPDH (CST, 2118), TREX1 (BD, 611986), PARP1 (CST, 9542) and H327kac (CST, 8173), Myc (MBL, 562), Tubulin (Sigma, T9026), Chk2 (CST, 2662), p-Chk2 (T68) (CST, 2197), ATM (abcam, ab32420), p-ATM (S1981) (abcam, ab81292). Membranes were stained using horseradish peroxidase conjugated secondary anti-rabbit antibody (CST, 7076 S) or anti-mouse antibody (CST, 7074 S). Blots were developed using SuperSignal West Pico PLUS substrate (Thermo Fisher) and scanned with a Bio-Rad XRS+ gel imaging system.

## Subcellular fractionation

BMDMs were harvested in ice-cold PBS supplemented with 1 mM EDTA on ice using a cell scraper and centrifuged at 300×*g* for 5 min at 4 °C. BMDMs were then resuspended in 5 mL complete DMEM and counted. Briefly, $3 \times 10^{6}$ cells per condition were taken for subcellular fractionation using the Subcellular fractionation kit for cultured cells (Thermo Fisher) according to the manufacturer's instructions. The protein content of membrane, cytosolic, soluble nuclear, and chromatin-bound nuclear extracts were quantitated via BCA assay (Thermo Fisher). Subcellular fractions were then resolved via SDS-PAGE and immunoblotted for proteins of interest using antibodies specific to individual subcellular compartments.

## Flow cytometry

To assess cellular viability, cell suspensions were washed with PBS to remove residual FBS then stained with Zombie NIR™ (BioLegend, 423106) in PBS for 15 min on ice. Cells were washed and fixed in 4% PFA for 10 min at room temperature. After fixation, cells were permeabilized in 90% methanol for 15 min on ice. Cells were washed to remove residual methanol, and stained for γH2AX (CST, 9718) and HA (CST, 2367) in FACS buffer for 1 h on ice. Fixed cells were then stained with fluorescently labeled secondary antibodies, AF488 donkey anti-rabbit IgG (Invitrogen, A-21206) and AF647 donkey anti-mouse IgG (Invitrogen, A-31571). For flow cytometry studies of splenocytes, spleens were mashed through a 70-µm strainer and washed in PBS. Red blood cells were lysed in ACK Lysing Buffer (Gibco, A10492-01) before staining with Zombie NIR in PBS for 15 min on ice. Cells were then stained for CD45 (BV605, BioLegend, 30-F11), CD4 (BV421, BioLegend, GK1.5), CD8a (PerCP/Cy5.5, BioLegend, 53-6.7), CD19 (FITC, BioLegend, 6D5), and NK1.1 (PE, BioLegend, PK136), or CD45 (BV605, BioLegend, 30-F11), CD11c (AF488, BioLegend, N418), CD11b (BV510, BioLegend, M1/70), MHCII (PE, BioLegend, M5/114.15.2), Ly6G (PerCP/Cy5.5, BioLegend 1A8), Ly6C(BV421, BioLegend, HK1.4), and F4/80 (AF700, BioLegend, BM8) in FACS buffer for 30 min on ice. For all experiments, Fc-

mediated interactions were blocked by incubating cell suspensions with purified rat anti-mouse CD16/32 (BD Biosciences, 553142) in FACS buffer during primary staining. In all flow cytometry experiments, cells were analyzed on an Attune NxT Flow Cytometer (Thermo Fisher) or LSR II (BD Biosciences) and data analysis was conducted in FlowJo™ v10 software (FlowJo LLC).

## Histology

Livers were fixed in 4% paraformaldehyde, washed with 70% ethanol prior to embedding in paraffin wax and sectioning. Three-μm tissue sections were stained with hemotoxylin and eosin (H&E) and slides were imaged on an EVOS M5000 microscope (Thermo Fisher).

## Immunofluorescence

IMR-90 cells and Flp-in 293 cells expressing myc-tagged TREX1 were cultivated on poly-L-lysine coated coverslips before fixation. BMDMs were cultivated on gelatin-coated coverslips before fixation with 4% paraformaldehyde for 15 min at room temperature and permeabilized with 0.3% Triton X-100 for 20 min at room temperature. Coverslips were washed with 1x PBS and blocked with blocking buffer (5% BSA in DPBS supplemented with 3% goat serum) for 60 min. Coverslips were immunostained for γH2AX (abcam, ab26350), 53BP1 (Novus Biologicals, NB100-904), pATM (Rockland, 200-301-400), ER-associated protein disulfide isomerase (PDI) (Thermo Fisher, S34253), myc (MBL, 562) and HA (CST, 2367 S) in blocking buffer for 60 min at 4 °C, and washed with PBS. Primary stains were visualized with the addition of anti-rabbit or anti-mouse fluorescent-conjugated secondary antibody. Nuclei were stained with 4′,6-Diamidino-2-phenylindole dihydrochloride (DAPI; Thermo Fisher) and mounted in ProLong™ gold anti-fade mounting medium (Thermo Fisher). Images were acquired on the Leica TCS SP8 WLL Confocal. Images were analyzed in ImageJ version 2.9.0.

## Gene expression analysis

Total RNA from MEFs, BMDMs and human PBMCs were extracted using the RNAeasy mini kit (Qiagen) according to manufacturer's instructions. Total RNAs were treated with RQ1-Rnase-free Dnase (Promega, M6101) and TaqMan RNA-to-Ct 1-step kit (Applied biosystems) was used to measure mRNA expression. Primer probe assays were obtained from Integrated DNA Technologies (IDT). Full sequence information and assay identification numbers are tabulated in sequence information (Supplementary Data 4). Ct values of target genes were normalized to the values of the reference gene *Polr2a*. Fold change in target gene expression was reported as $2^{-\Delta\Delta Ct}$, normalized to the average expression observed in WT mice.

## RNA sequencing

RNA of IMR-90 cells stably expressing RVCL TREX1 (V235Gfs) by doxycycline for 2 weeks and in cells without doxycycline was extracted using Direct-zol RNA Miniprep (Zymo Research). The RNA-seq library was prepared using an Illumina TruSeq Stranded mRNA Sample Prep Kit according to the manufacturer's instructions. The libraries were pooled and sequenced on the Illumina NovaSeq 6000 platform (high output mode, 2 × 100-base paired-end). The clean reads were mapped to the human reference genome sequence (GRCm38/p13.genome). Deseq2 was used to identify differentially expressed genes with an adjusted *P*-value < 0.05. Gene set enrichment analysis was performed using the GSEA software (Broad Institute) using SASP gene panel[89]. These RNA-seq raw data using IMR90 are available from the DDBJ Sequence Read Archive (https://ddbj.nig.ac.jp/DRASearch/ – accession numbers DRA016748).

## ADP-lite cell viability assay

WT and heterozygous RVCL MEFs were seeded at 1,000 cells per well and treated with indicated concentrations of PARP inhibitors, olaparib or talazoparib, for 72 and 96 h. Viability of treated cells was assessed using ADP-lite cell viability assay (PerkinElmer) according to the manufacturer's instructions. ATP-lite assay and drug treatments of cells was performed using a D300e liquid handling system (TECAN) by the High-throughput Screening Core (University of Pennsylvania).

## CellTiter-glo viability assay

WT and RVCL MEFs were seeded at 1500 cells per well and treated with 100 IU mouse IFN-β (PBL Assay Science, 12400) for 24 h. Following pre-treatment MEFs were treated with indicated concentrations of aclarubicin for 24 h. Viability of treated cells as assessed using Cell-titer glo luminescent cell viability assay (Promega) according to the manufacturer's instructions.

## Sensitive Identification of Individual DNA Ends (STRIDE) analysis

dSTRIDE were performed as described previously[37] by intoDNA. Briefly, BMDMs from LysM-Cre-positive WT TREX1 and LysM-Cre-positive RVCL mutant TREX1 micewere dissociated on Day 7 post-differentiation and seeded into gelatin-coated glass cover slips. On day 8, BMDMs were treated with 10 μM olaparib or DMSO for 72 h, refreshing drug media every 24 h. Cells were fixed with ice-cold 70% ethanol and shipped to intoDNA for analysis. TREX1 expression was assessed by staining with anti-HA conjugated antibody.

## TREX1 inhibitor synthesis

The TREX1 inhibitor Compound 16 was synthesized as previously described[70]. All solvents and chemicals were purchased from commercially available sources and used without further purification, or purified according to Purification of Laboratory Chemicals[90]. Solvents were dried under standard conditions. Reactions were monitored by thin layer chromatography (TLC) using pre-coated silica on aluminum plates from Merck (60F$_{254}$). TLC plates were visualized with ultraviolet light and/or by treatment with ceric ammonium molybdate solution (CAM) and heating. Products were purified on column chromatography with Silica gel 60 from Macherey Nagel (0.036–0.071 mm; 215–400 mesh), a CombiFlash Rf+ Teledyne Isco system fitted with pre-packed silica gel columns (Interchim) or/and preparative HPLC Quaternary Gradient 2545 equipped with a Photodiode Array detector (Waters) fitted with a reverse phase column (Xbridge Prep C18 5 μm OBD, 30 × 150 mm). NMR spectroscopy was performed on Bruker spectrometers to confirm Compound 16 was in accordance with the previous spectra described by Letourneau et al.[70]. The purity of final compounds, determined to be >95% by UPLC MS, was recorded on a Waters Acquity H-class equipped with a Photodiode array detector and SQ Detector 2 with a reverse phase column (Aquity UPLC® BEH C18 1.7 μm, 2.1 × 50 mm).

## Inhibitor and interferon treatments

BMDMs (Day 7 of culture) or MEFs were treated with 10 μM olaparib in complete DMEM for 72 h, where the media was refreshed every 24 h with fresh olaparib. For TREX1 inhibitor experiments, BMDMs were treated with 10 μM TREX1 inhibitor with or without 10 μM olaparib for 72 h. Media was refreshed every 24 h with fresh TREX1 inhibitor and olaparib. For CPT treatment, BMDMs were incubated with 1.25 μM CPT in complete DMEM for 48 h. For interferon treatments, BMDMs were treated with 100 IU/mL of IFN-β (PBL Assay Science, 12400) or 10 μg/mL of IFN-γ (Peprotech, 315-05) for 72 h. Media was refreshed every 24 h with fresh IFN-β or IFN-γ. Following incubation, BMDMs were mechanically disassociated from the culture surface with 1 μM EDTA in PBS, while MEFs were trypsinized in 0.25% trypsin for 5 min to remove from the culture surface. Cell suspensions were then analyzed by flow cytometry.

## PacBio HiFi long-read and Illumina short-read sequencing of CRISPR-Cas9 cleavage site

Modifications to CRISPR/Cas9-induced DSB repair by wild-type or RVCL mutant TREX1 expression were analyzed by PacBio HiFi Long-

Read sequencing and Illumina short-read sequencing. HiFi long-read sequencing used the SMRT approach for targeted amplicon sequencing, with an initial round of PCR on extracted genomic DNA as a template to produce a 3797 bp fragment containing the CRSPR/Cas9 cleavage site using 5′-end AmMC6 modified primers (hACTB_long_F: /5AmMC6/GTAAAACGACGGCCAGT-ACCCTGAAGTACCCCATCGA, hACTB_long_R: /5AmMC6/CAGGAAACAGCTATGAC-GGACACGGAACA CATCTGGT) with Q5 High-Fidelity 2X Master Mix (New England Biolabs), followed by a second PCR step using barcoded (Index) universal primers (Index-F: /5phos/GGTAG-Index1-GTAAAACGACGGCCAGT, Index-R: 2nd Reverse/5phos/GGTAG-Index2-CAGGAAACAGCTATGAC). Barcoded fragments from each sample were pooled in equimolar amounts and purified using AMPure PB beads. SMRTbell libraries were then prepared using the Template Preparation Kit (Pacific Biosciences) according to the manufacturer's instructions (https://www.protocols. io/view/smrt-ots-bjugkntw). Sequencing was performed on the PacBio Sequel IIe system using Sequel II binding kit 2.2. These long-read sequence raw data are available from the DDBJ (https://www.ddbj.nig. ac.jp/index.html – accession number PRJDB17735). Using SMRT Link (ver. 13.0.0.207600), overhang adapter sequences were removed from the sequenced sequences to form sub-reads. After creating a consensus sequence (CCS) aligned with these subreads, CCS reads with an average quality value of less than 20 per read were removed and designated as HiFi reads. Using lima (ver. 2.7.1), reads were sorted by index, and adapter and index sequences in primers were removed. The obtained read sequences were mapped to a reference sequence using pbmm2 (ver. 1.13.0). Among the mapped sequences, reads holding 20 bp at both the 5′ and 3′ ends of the PCR-amplified sequence were extracted, and the frequency of the final extracted reads was tabulated by deletion size within each sequence.

The frequency of short deletions (<50 bp) was analyzed by amplicon sequencing using the Miseq. Genomic DNA surrounding the CRISPR/Cas9 target site was amplified using barcoded primers (hACTB_short_F: ACACTCTTTCCCTACACGACGCTCTTCCGATCTGT CACATCCAGGGTCCTCAC, hACTB_short_R: GTGACTGGAGTTCAG ACGTGTGCTCTTCCGATCTAGAGAAGTGGGGTGGCTTTT). Illumina index sequences were added by a second round of PCR using index-primers (2nd F: AATGATACGGCGACCACCGAGATCTACAC ACACTCTTTCCCTACACGACGC, 2nd R: CAAGCAGAAGACGGCAT ACGAGAT Index1 GTGACTGGAGTTCAGACGTGTG). Purified final libraries using VAHTS DNA Clean Beads (Vazyme) were sequenced using MiSeq. These long-read sequence raw data are available from the DDBJ (https://www.ddbj.nig.ac.jp/index.html – accession number PRJDB17744). Mutagenic alleles were called using CRISPResso2[91], and called reads with short deletions ( < 50 bp) were manually curated.

## Mass spectrometry

For the identification of cellular interacting proteins of WT TREX1 and TREX1 V235Gfs, large-scale FLAG pulldown was performed by transfecting 12 × 10 cm dishes of HEK 293 T cells with 24 μg of DNA per dish. Cells were transfected with WT TREX1 or TREX1 V235Gfs in the pcDNA 3.1 (-) vector containing an N-terminal 3x-FLAG tag using NheI and HindIII restriction sites. Empty vector-transfected cells were included as a control. Cells were harvested and lysed in ice-cold Nonidet P-40 (NP-40) buffer (50 mM HEPES [pH 7.4], 150 mM NaCl, 1% (v/v) NP-40, 1:100 protease inhibitor cocktail, 1:100 phosphatase inhibitor cocktail). Whole cell lysates (WCLs) were initially cleared by centrifugation at 21,000×$g$ for 20 min at 4 °C, then further pre-cleared using Sepharose beads (Sigma, catalog no 4B200) for 2 h at 4 °C. Precipitates were then incubated with anti-FLAG M2 (Sigma, catalog no M8823) agarose beads at 4 °C for 4 h, extensively washed with 1% NP-40 lysis buffer, and separated on a NuPAGE 4 to 12% Bis-Tris gradient gel (ThermoFisher, catalog no NP0324). The protein gel was silver stained according to the manufacturer's instructions (Invitrogen SilverQuest Silver Staining Ki, catalog no LC6070), and the bands specific to WT TREX1 and TREX1

V235Gfs (but not present in the vector control) were excised and analyzed by ion-trap mass spectrometry at the Harvard Taplin Biological Mass Spectrometry Facility in Boston, MA. The mass spectrometry proteomics data are available from the ProteomeXchange Consortium (https://www.proteomexchange.org) via the PRIDE partner repository with the accession number PXD051905.

## Human cytokine analysis

Sera from healthy controls and patients was collected and submitted for cytokine analysis performed by Eve Technologies. Samples were anonymous, genetically confirmed, and lacked identifying information or associated clinical data. Sample collection was exempt from IRB since samples were anonymous, not tied to clinical data, and therefore not categorized as human subjects research.

## Human embryo crossover analysis

Trophectoderm samples from embryos biopsied at day 5–7 of development underwent whole genome amplification using multiple displacement amplification (RepliG, Qiagen, Germany) and were processed in comparison to family control DNA on the HumanKaryomap-12 Array (Illumina, United States). Genome-wide haplotyping was performed using BlueFuse Multi Software (Illumina, United States) and crossover events identified as switching in the phase of key and non-key SNPs across each maternal chromosome. The frequency of crossover events in embryos of an RVCL patient was compared to the publicly available results of Hou et al.[56], who quantitated crossover events using SNP loci of phased haploid cells and a hidden Markov model to infer crossover events. The clinical results including SNP and deletion analysis arose as part of routine, standard-of-care clinical testing. No experiments, other genetic analyses or sequencing were performed on human embryos.

## Epidemiological analysis

The odds ratio (OR) of breast cancer was calculated by dividing the odds of breast cancer in female patients with RVCL (3:16) by the odds of breast cancer in women under age 50 years in the general population of the US (2100:97900) from the publicly available SEER*Explorer dataset[92]. The OR of breast cancer in RVCL was compared to published ORs before age 50 years in *BRCA1* and *BRCA2*[93].

## Statistics

Statistical analyses are indicated in the figure legends and were conducted in Prism (GraphPad) Version 9.5.1.

## Reporting summary

Further information on research design is available in the Nature Portfolio Reporting Summary linked to this article.

# Data availability

RNA-seq data are available at the DDBJ Sequence Read Archive (https:// ddbj.nig.ac.jp) with the accession number DRA016748. Mass spectrometry proteomics data are available via the ProteomeXchange Consortium (https://www.proteomexchange.org) with the identifier PXD051905. Source data are provided with this paper.

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

## Acknowledgements

We acknowledge Michael S. Diamond and John P. Atkinson (WashU) for facilitating Dr. Miner's goal of generating the TREX1 T235Gfs mice during his postdoctoral fellowship in preparation for his independent career. We thank the late Herbert C. Morse III as well as Alexander L. Kovalchuk for giving us TREX1-dsRed reporter mice. We also thank Andria Ford, Alexander L. Kovalchuk and Nilufar Rahimova (helpful discussions), Alexandra Lopez (pilot experiments), Adam Zuiani and Canyu Liu (critical reading of the manuscript), as well as Sally Thompson, Ashley Batista, and Rennie Rhee (REVEAL study). Additionally, we thank the late Michelle Noll for managing the TREX1 T235Gfs mouse colony as it was initially being generated. This work was supported by National Institutes of Health grant K08AR070918 (J.J.M.), National Institutes of Health grant R01AI143982 (J.J.M.), National Institutes of Health grant R01NS131480 (J.J.M.), Scientist Development Award from the Rheumatology Research Foundation (J.J.M.), Penn Colton Center for Autoimmunity pilot award (J.J.M.), gift from the Clayco Foundation (J.J.M., M.U.G., and N.M.), Penn RVCL Sisters Fund (J.J.M.), National Institutes of Health Medical Scientist Training Program T32 GM007170 (S.D.C.), Japan Agency for Medical Research and Development grant 22ek0109424h0003 (O.O.), Japan Society for the Promotion of Science grant 22H00466 (O.O. and T.K.), Japan Society for the Promotion of Science grant 22H02981 (T.K.), Japan Society for the Promotion of Science grant 18K07522 (T.K.), and Japan MHLW for Research on Intractable Disease grant JP21FC1007 (O.O.).

## Author contributions

Conceptualization: S.Chauvin, S.A., O.O., T.K., and J.J.M. Methodology: O.O., T.K., and J.J.M.; Validation: S.Chauvin, S.A., J.A.H., S.P., C.A.M., O.O., T.K., and J.J.M. Formal analysis: S.Krishnamurthy and M.N.A. Investigation: S.Chauvin, S.A., J.A.H., S.P., A.S., Y.Nitta, R.K., Y.H., R.S., S.Kitahara, S.Koide, A.K., J.F., C.A.M., W.A.S., W.Q., N.S., S.Krishnamurthy, F.R.Z., Y.Ning, L.K., O.P., P.S.A., C.C., G.L., L.C., R.R., N.M., M.N.A., D.C.S., P.T.C., M.U.G., T.H., O.O., T.K., and J.J.M. Resources: O.O., T.K., and J.J.M. Data curation: S.Chauvin, S.A., J.A.H., S.P., O.O., T.K., and J.J.M. Writing – original draft: S.Chauvin, J.A.H., S.P., and J.J.M. Writing – review and editing: S.Chauvin, J.A.H., N.M., P.T.C., E.D.O.R., Y.B., R.A.G., S.Cherry, M.U.G., T.H., O.O., T.K., and J.J.M. Visualization: S.Chauvin, S.A., J.A.H., F.R.Z., O.O., T.K., and J.J.M. Supervision: N.M., P.T.C., E.D.O.R., Y.B., R.A.G., S.Cherry, M.U.G., T.H., O.O., T.K., and J.J.M. Project administration: O.O., T.K., and J.J.M. Funding acquisition: O.O., T.K., and J.J.M.

## Competing interests

The authors declare no competing interests.

## Additional information

**Samuel D. Chauvin** [1,2,26], **Shoichiro Ando** [3,26], **Joe A. Holley** [1,2,26], **Atsushi Sugie** [4], **Fang R. Zhao** [5], **Subhajit Poddar** [1,2], **Rei Kato** [3], **Cathrine A. Miner** [1,2], **Yohei Nitta** [4], **Siddharth R. Krishnamurthy** [6,7], **Rie Saito** [8], **Yue Ning** [1,2], **Yuya Hatano** [3], **Sho Kitahara** [3], **Shin Koide** [3], **W. Alexander Stinson** [5], **Jiayuan Fu** [1,2], **Nehalee Surve** [1,2], **Lindsay Kumble** [1,2], **Wei Qian** [5], **Oleksiy Polishchuk** [1,2], **Prabhakar S. Andhey** [9], **Cindy Chiang** [10,25], **Guanqun Liu** [10,25], **Ludovic Colombeau** [11], **Raphaël Rodriguez** [11], **Nicolas Manel** [12], **Akiyoshi Kakita** [8], **Maxim N. Artyomov** [9], **David C. Schultz** [13], **P. Toby Coates** [14,15], **Elisha D. O. Roberson** [5], **Yasmine Belkaid** [6,7,16], **Roger A. Greenberg** [17], **Sara Cherry** [18,19], **Michaela U. Gack** [10,25], **Tristan Hardy** [20,21], **Osamu Onodera** [3,22], **Taisuke Kato** [22] ✉ & **Jonathan J. Miner** [1,2,5,18,23,24] ✉

[1]Division of Rheumatology, Department of Medicine, University of Pennsylvania Perelman School of Medicine, Philadelphia, PA, USA. [2]RVCL Research Center, University of Pennsylvania Perelman School of Medicine, Philadelphia, PA, USA. [3]Department of Neurology, Clinical Neuroscience Branch, Brain Research Institute, Niigata University, Niigata, Japan. [4]Department of Neuroscience of Disease, Brain Research Institute, Niigata University, Niigata, Japan. [5]Department of Medicine, Washington University in Saint Louis, Saint Louis, MO, USA. [6]Metaorganism Immunity Section, Laboratory of Immune System Biology, National Institute of Allergy and Infectious Diseases, National Institutes of Health, Bethesda, MD, USA. [7]NIAID Microbiome Program, National Institute of Allergy and Infectious Diseases, National Institutes of Health, Bethesda, MD, USA. [8]Department of Pathology, Clinical Neuroscience Branch, Brain Research Institute, Niigata University, Niigata, Japan. [9]Department of Pathology and Immunology, Washington University in Saint Louis, Saint Louis, MO, USA. [10]Department of Microbiology, The University of Chicago, Chicago, IL, USA. [11]Equipe Labellisée Ligue Contre le Cancer, Institut Curie, CNRS, INSERM, PSL Research University, Paris, France. [12]INSERM U932, Institut Curie, PSL Research University, Paris, France. [13]High-throughput Screening Core, University of Pennsylvania,

Philadelphia, PA, USA. [14]Central and Northern Adelaide Renal and Transplantation Service (CNARTS), The Royal Adelaide Hospital, Adelaide, South Australia, Australia. [15]School of Medicine, Faculty of Health Sciences, University of Adelaide, Adelaide, South Australia, Australia. [16]Institut Pasteur, Paris, France. [17]Department of Cancer Biology, Penn Center for Genome Integrity, Basser Center for BRCA, Perelman School of Medicine, University of Pennsylvania, Philadelphia, PA, USA. [18]Institute for Immunology and Immune Health, University of Pennsylvania Perelman School of Medicine, Philadelphia, PA, USA. [19]Department of Pathology and Laboratory Medicine, University of Pennsylvania Perelman School of Medicine, Philadelphia, PA, USA. [20]Genetics, Repromed, Monash IVF, Dulwich, South Australia, Australia. [21]Genetics and Molecular Pathology, SA Pathology, Adelaide, Australia. [22]Department of Molecular Neuroscience, Brain Science Branch, Brain Research Institute, Niigata University, Niigata, Japan. [23]Department of Microbiology, University of Pennsylvania Perelman School of Medicine, Philadelphia, PA, USA. [24]Penn Colton Center for Autoimmunity, University of Pennsylvania Perelman School of Medicine, Philadelphia, PA, USA. [25]Present address: Florida Research and Innovation Center, Cleveland Clinic, Port Saint Lucie, FL, USA. [26]These authors contributed equally: Samuel D. Chauvin, Shoichiro Ando, Joe A. Holley. ✉e-mail: taisuke8077@bri.niigata-u.ac.jp; Jonathan.Miner@PennMedicine.UPenn.edu

