## [Peer Review File · Nature Communications]

Inherited C-terminal TREX1 variants disrupt homology-directed repair to cause senescence and DNA damage phenotypes in *Drosophila*, mice, and humansREVIEWER COMMENTS

Reviewer #1 (Remarks to the Author):

This manuscript from Holley et al investigates mechanisms of the inherited retinal vasculopathy with cerebral leukoencephalopathy and systemic manifestations (RVCL) disease. RVCL is caused by expression of a truncated form of the exonuclease TREX1 which disrupts its normal association with the ER. RVCL TREX1 protein displays pancellular localization including access to the nucleus. In contrast to TREX1 mutations linked to severe immune diseases like Aicardi-Goutieres Syndrome (AGS), these mutations preserve TREX1 catalytic activity. Of further distinction from AGS, where TREX1 is often biallelically mutated, RVCL is caused by dominant mutations in TREX1. Therefore, AGS and RVCL can both be caused by mutations in TREX1 but exhibit distinct etiologies and manifestations. Indeed, AGS has been classified as a Type I interferonopathy whereas RVCL has previously been demonstrated to lack an IFN signature. Here, Holley et al propose that expression of TREX1 mutants that lack ER association causes RVCL by inhibiting nuclear homologous recombination. This is an interesting and plausible proposal that is incompletely supported by the data presented.

One major concern I have regarding the manuscript is that many key results derive from the HEK293 FlpIn system that allows for conditional expression of TREX1. Use of the HEK293 FlpIn line may result in TREX1 overexpression to a degree far beyond what is normally observed - even in aged tissues. It is critical that the authors compare TREX1 levels achieved using this system to control lines. My major concern is that high levels of TREX1 expression are leading to potential artifacts.

Recapitulation of some results in genetically engineered mouse cell lines partially mitigated the above concern, however many of these experiments were also based on transgenic overexpression of human TREX1 RVCL from the ROSA locus potentially resulting in overexpression.

Related to the point above TREX1 overexpression (w/o RVCL mutation) is often sufficient to drive substantial RVCL phenotypes (i.e. Fig. 1a,b; Fig. 5b). However, as expected, full length TREX1 localization appears to be restricted to the ER/cytosol and excluded from the nucleus in the images shown (i.e. Fig. 1c; Extended Data Fig. 1a). These findings seem to be at odds with the central model of the manuscript that TREX1 RVCL-dependent DNA damage derives from its nuclear access.

The authors propose that TREX1-RVCL causes disease by blocking HR. I was therefore surprised by the finding that RAD50 scored as one of the strongest suppressors of the rough eye phenotype in flies (Fig. 1e), given its well-established role in promoting HR. Perhaps the authors can more clearly articulate how these findings fit within the larger model for non-experts in DNA repair.

The authors find that RVCL TREX1 causes increased DNA damage in primary cells treated with PARP inhibitors (Fig. 3). The interpretation is that this reflects deficient repair upon RVCL TREX1 expression. However, it is difficult to understand how these results apply to more physiological settings where RVCL progresses in the absence of olaparib treatment. One possibility - consistent with aspects of the model proposed - is that defects accrue over time to explain the long latency of RVCL.

Related to the point above the proposal of organ-specific TREX1 induction in aged individuals is not well-supported by the data.

Treatment with a TREX1 small molecule inhibitor reverses DNA damage phenotypes in primary cells (Fig. 3g). However, TREX1 catalytic inhibitors have not been extensively validated leaving their potency and potential for indirect effects to be important, open questions. While I appreciate that the genetic tools used elsewhere in the manuscript are difficult to apply in this setting, I am concerned about the rigor of this particular experiment.

The authors spend considerable effort to investigate the role of interferon in RVCL based on the

premise that previous studies have suggested that TREX1 is caused by loss of TREX1 function. This is not a fair characterization of the field where numerous studies have demonstrated the functionality of C-terminally truncated TREX1 and the lack of IFN in RVCL (for instance see PMID: 33476576, 28420733). Even the directly cited studies (PMID: 27230542, 26320659) are taken somewhat out of context as they focus on proposed cGAS-independent roles for TREX1 including regulation of oligosaccharyltransferase activity.

Reviewer #2 (Remarks to the Author):

In this study, Holley et al propose and provide evidence for a DNA damage-based theory that elucidates the mechanism underlying the clinical phenotypes of TREX1 mutations associated with retinal vasculopathy with cerebral leukoencephalopathy (RVCL). Using a range of model systems including *Drosophila*, mice and human cell culture and a variety of methods and assays, the authors show that RVCL-associated TREX1 mutants are mislocalized in the nucleus, which subsequently causes intrinsic DNA damage by disrupting homology-mediated repair processes.

The authors present key findings to support their hypothesis: 1) The Detection of DNA damage and DNA damage response in cells expressing the mutants, contingent on TREX1's nuclease activity and nuclear localization; 2) The rescue of the rough eye phenotype in *Drosophila* caused by a RVCL-associated human TREX1 mutant through the depletion of the genes involved in the DNA damage response; 3) The detection of senescence in RVCL TREX1-expressing human cells, which, again, is dependent on TREX1 nuclease activity and its nuclear localization; 4) The observation of DNA damage and cell death (mild) caused by RVCL TREX1 nuclease activity in the presence of PARPi; 5) The detection of inhibitory effects of RVCL TREX1 on HR, which also hinge on TREX1's nuclease activity and nuclear localization; 6) The observation of aneuploidy, reduced meiotic crossover and breast cancer incidents in female RVCL patients; and 7) The observation of increased sensitivity of RVCL TREX1 expressing cells to aclarubicin (although only at high concentrations) and more severe brain lesions in an RVCL patient following aclarubicin treatment. The study further shows that RVCL mutations, contrary to mutations that cause AGS and other autoimmune disorders, do not trigger interferon production in the presence or absence of exogenous cytosolic DNA stimuli, indicating a unique pathway in RVCL pathogenesis.

The results described in the manuscript provide intriguing insights into RVCL, highlighting distinct impacts of TREX1 mutations, with important implications for the treatment of RVCL. The strategic use of multiple model systems and the comprehensive examination of the RVCL TREX1's role in a range of DNA damage-related cellular processes constitute the core strengths of the study. However, to ensure rigor the study would benefit from a deeper exploration of the molecular mechanisms, and there are several concerns that need to be addressed.

Points:

1. It is unclear exactly how RVCL TREX1 mutants interfere with DNA repair (especially HR) and cause DNA damage.
2. The authors show that RVCL TREX1 expression suppressed HR and enhanced NHEJ in human cells. They also show that RAD50 depletion apparently mitigated the RVCL TREX1-induced rough eye phenotype in *Drosophila*. It is hard to reconcile these observations, considering the established role of RAD50 in facilitating HR.
3. It is also hard to reconcile the discrepancy between the very mild PARPi sensitivity depicted in Extended Data Figure 5a and the pronounced suppression of HR repair shown in Figure 5b and 5c. The authors should compare the effects of RVCL TREX1 with BRCA1/2 deficiency on HR and PARPi sensitivity.
4. The authors suggest an interesting hypothesis that RVCL TREX1 may digest the 3' overhang of the resected DSB ends, thereby impeding HR. They should examine whether there is a sequence loss at

the HR reporter cutting site in Figure 5a in cells expressing RVCL TREX1.

Reviewer #3 (Remarks to the Author):

Holley et al, investigate how mislocalization of the intracellular DNase Trex1 causes small vessel disease. Trex1 is usually anchored in the ER with its catalytic center facing the cytosol. Loss of its nuclease activity causes cGAS/STING dependent systemic autoimmunity. Similar mechanisms have been suggested for c-terminal truncation mutants (lacking its membrane anchor) that cause RVCL. However, a deep understanding of the disease pathogenesis is lacking. The authors undertook a comprehensive multi-species approach to gain insights into mechanisms that could drive pathogenesis and show that DNase active Trex1 RVCL mutants interfere with DNA DSB repair. As such, RVCL TREX1 mutants sensitize to DNA damage-inducing therapies, which is contraindicated in RVCL based on the current study.

The manuscript presents extensive data that is coherent between many models and species. The study adds a lot of new information on the previously identified nuclear TREX1 with real world implications for the treatment of RVCL and cautions in DNA damage inducing therapies outside of the cancer treatment.

I only have a few points that should be addressed:

Overexpression of WT Trex1: In Figure 1, how does human WT Trex1 cause the rough eye phenotype if it's not mislocalized as suggested by the IF-pictures throughout the manuscript? On the contrary: The quantification of IF in Figure 1D suggest that a substantial fraction of Trex1 WT is nuclear (mean ratio nuclear/cyto approx 0.75). Can the authors clarify this also in regards of using Trex1 WT overexpression in later experiments and not observing any effects? How do the expression levels of TREX1 in their systems relate to endogenous levels of TREX1 or RVCL TREX1? In Figure 2H the text says that RVCL TREX1 did not colocalize with SET, in the legend it says myc-TREX1. Was the experiment done with mutant or WT TREX1. If it was done with mutant, would their WT TREX1 colocalize with the SET complex?

Figure 4D: the Ifna2 levels in the HC group seems quite high. SIMOA study showed steady state levels of Ifna in the fg/ml range, here it's one log higher. Also, RVCL patients have been screened previously for IFN and showed no difference in SIMOA having Ifna in fg/ml range (PMID: 28420733). The amounts detected here, would be consistent with an SLE or AGS diagnosis in the HC group. This assay should be checked carefully.

Blocking of HDR: The authors use a CRISPR-bases system with a ssDNA donor to study repair efficiency in cells expressing Trex1 variants and conclude, that HDR is impaired in the presence of RVCL TREX1. If I am not mistaken, this setup introduces genetic alterations via a mechanism that is termed single strand template repair involving the Fanconi anemia pathway and is distinct from HDR (PMID: 30054595). In fact, RAD50 regulates FANCD2 activity (PMID: 19609304). Can the authors provide more mechanistic insights into the exact molecular mechanism underling the repair defect and potentially adjust the manuscript accordingly? e.g. does TREX1 colocalize / compete with Faconi factors?

Some of the authors published seminal papers in establishing the connection between DNA damage and activation interferon via cGAS/STING. Can the authors clarify and speculate why the DNA damage caused by TREX1 RVCL (Fig. 2 and related) is not immunogenic? Also, the authors cite work on TREX1 translocation into the nucleus (Nader et al., lead author is co-author here), which causes invasive

reprogramming of cancer cells. Do the RVCL cells show similar reprogramming? This could be an alternative mechanism underlying carcinogenesis.

We thank the reviewers for the positive assessment of our manuscript.

- In the revised version of the manuscript, we have performed numerous additional experiments to address reviewer questions.*
- We merged our manuscript with a second manuscript. We felt that merging the two manuscripts was necessary to fully address the questions about TREX1 over-expression.*
- Now, three different authors (including 2 trainees) will be sharing first-authorship of one manuscript.*

In this revision, we include:

- Four new figures of original data.*
- New data showing the interactions of cytokines, DNA damage, and TREX1 expression (**Figure 4**).*
- Newly generated TREX1-dsRed reporter mice used to show which cell types have high endogenous TREX1 expression—under control of the endogenous promoter (**Figure 5**).*
- Mouse data showing cytokine-induced reduction of specific immune cell subsets that express TREX1 (**Figure 6**).*
- New data showing that RVCL mutant TREX1 interacts with chromatin (**Figure 8**).*
- **To address concerns about over-expression, all of these additional studies were performed in mice that express TREX1 only under control of the endogenous promoter, without over-expression.***

Please see our point-by-point response below.

REVIEWER COMMENTS

Reviewer #1 (Remarks to the Author):

This manuscript from Holley et al investigates mechanisms of the inherited retinal vasculopathy with cerebral leukoencephalopathy and systemic manifestations (RVCL) disease. RVCL is caused by expression of a truncated form of the exonuclease TREX1 which disrupts its normal association with the ER. RVCL TREX1 protein displays pan-cellular localization including access to the nucleus. In contrast to TREX1 mutations linked to severe immune diseases like Aicardi-Goutieres Syndrome (AGS), these mutations preserve TREX1 catalytic activity. Of further distinction from AGS, where TREX1 is often biallelically mutated, RVCL is caused by dominant mutations in TREX1. Therefore, AGS and RVCL can both be caused by mutations in TREX1 but exhibit distinct etiologies and manifestations. Indeed, AGS has been classified as a Type I interferonopathy whereas RVCL has previously been demonstrated to lack an IFN signature. Here, Holley et al propose that expression of TREX1 mutants that lack ER association causes RVCL by inhibiting nuclear homologous recombination. This is an interesting and plausible proposal that is incompletely supported by the data presented.

We thank the Reviewer for positive assessment of our manuscript. We agree with the concerns about over-expression. In the revised manuscript, we added multiple new experiments and figures to address these concerns.

One major concern I have regarding the manuscript is that many key results derive from the HEK293 FlpIn system that allows for conditional expression of TREX1. Use of the HEK293 FlpIn line may result in TREX1 overexpression to a degree far beyond what is normally observed - even in aged tissues. It is critical that the authors compare TREX1 levels achieved using this system to control lines. My major concern is that high levels of TREX1 expression are leading to potential artifacts.

We agree that over-expression is a limitation of the HEK 293 cell experiments.

In the revised manuscript, we acknowledge this and include several additional experiments demonstrating that the TREX1 mutant causes DNA damage in primary cells even when expressed from the endogenous

locus, under control of the endogenous promoter. These experiments utilize primary bone marrow-derived macrophages (BMDMs). **Please see Figures 4 and 6 of the revised manuscript.**

We used the Flp-In single-site integration system because higher expression levels of the mutant TREX1 are not tolerated. **Please see Reviewer Figure 1.** Single-site expression was necessary because the RVCL mutant TREX1 is toxic when expressed at high levels. We agree that the HEK 293 cell data were a limitation of our initial submission. This is why our revised submission includes many additional studies of TREX1 expressed under control of the endogenous promoter. Our revised manuscript includes extensive studies of DNA damage induced upon up-regulation of TREX1 by cytokines. **Please see Figures 4 and 6 of the revised manuscript.**

Recapitulation of some results in genetically engineered mouse cell lines partially mitigated the above concern, however many of these experiments were also based on transgenic overexpression of human TREX1 RVCL from the ROSA locus potentially resulting in overexpression.

We agree with the concerns about over-expression data in the first submission of our manuscript, so we have merged the initial submission with a second manuscript focusing on TREX1 expression under control of the endogenous promoter.

In the combined, revised manuscript, we demonstrate that up-regulation of endogenous TREX1 by cytokines is associated with DNA damage. We include several new figures focused on the DNA-damaging effects of the TREX1 mutant expressed at the endogenous locus, under control of the endogenous promoter. **Please see Figures 4, 5, and 6.**

In **Figure 4**, we show the relationship among inflammation, aging, DNA damage, and TREX1 expression in mice and in bone marrow-derived macrophages (BMDMs).

In **Figure 5**, we use newly generated TREX1-dsRed reporter mice to demonstrate that TREX1 expression levels are high in monocytes, macrophages, and CD11c⁺ leukocytes.

In **Figure 6**, we include data demonstrating that inflammation triggers up-regulation of TREX1 in mice, and that inflammation in RVCL mice causes loss of monocytes, macrophages, and CD11c⁺ cells within one week after the inflammatory insult. All of this work was performed in mice expressing RVCL mutant TREX1 under control of the endogenous promoter.

Thus, up-regulation of the TREX1 mutant at the endogenous locus is associated with DNA damage and loss of specific cell types that highly express TREX1.

Related to the point above TREX1 overexpression (w/o RVCL mutation) is often sufficient to drive substantial RVCL phenotypes (i.e. Fig. 1a,b; Fig. 5b). However, as expected, full length TREX1 localization appears to be restricted to the ER/cytosol and excluded from the nucleus in the images shown (i.e. Fig. 1c; Extended

Data Fig. 1a). These findings seem to be at odds with the central model of the manuscript that TREX1 RVCL-dependent DNA damage derives from its nuclear access.

We thank the Reviewer for pointing out that WT TREX1 over-expression can partially phenocopy the effect of RVCL mutant TREX1 in certain conditions. We agree. In the revised manuscript, we now discuss that over-expression of TREX1 may lead to aberrant activity, including some nuclear localization. Please see lines 151-152 of the revised manuscript.

The over-expression issue in Drosophila is a legitimate concern, which we now discuss on lines 475-478 of the revised manuscript.

The Drosophila work was primarily for hypothesis-generation, and we hope that questions about the results in Drosophila will be mitigated by our new mouse model studies of TREX1 mutant-associated DNA damage occurring when TREX1 is expressed under control of the endogenous promoter. Please see Figures 4 and 6.

The authors propose that TREX1-RVCL causes disease by blocking HR. I was therefore surprised by the finding that RAD50 scored as one of the strongest suppressors of the rough eye phenotype in flies (Fig. 1e), given its well-established role in promoting HR. Perhaps the authors can more clearly articulate how these findings fit within the larger model for non-experts in DNA repair.

We agree with the Reviewer that the Rad50 finding is initially perplexing given its role in promoting HDR. This can be explained by differences between Rad50 knockout and knockdown. Indeed, Rad50 knockout is pupal lethal in Drosophila¹ and embryonic lethal in mice².

However, unlike deletion or knockout of Rad50, partial down-regulation of Rad50 can be protective in the context of DNA damage. For example, Rimkus et. al. found that haploinsufficiency of Rad50 protected from the rough eye phenotype associated with ATM knockdown in Drosophila³. Additionally, Tuxworth et. al. showed that Rad50 heterozygosity in Drosophila could partially preserve climbing scores and survival of pigmented retinal cells in the A β ₁₋₄₂ model of dementia, which is triggered by inducing DNA double-strand breaks⁴. Thus, there are multiple examples in which Rad50 expression can be protective in the context of a DNA damage phenotype, including the rough eye phenotype associated with ATM knockdown.

Therefore, by using siRNA to only partially reduce Rad50 expression in Drosophila, we also protected against the damaging effects of RVCL mutant TREX1 by reducing DDR signaling. This is consistent with prior literature on the role of Rad50 expression levels in DNA damage-associated pathology. We now discuss this precedent and the distinction between Rad50 deletion and Rad50 knockdown. Please see lines 161-164 of the revised manuscript.

The authors find that RVCL TREX1 causes increased DNA damage in primary cells treated with PARP inhibitors (Fig. 3). The interpretation is that this reflects deficient repair upon RVCL TREX1 expression. However, it is difficult to understand how these results apply to more physiological settings where RVCL progresses in the absence of olaparib treatment. One possibility - consistent with aspects of the model proposed - is that defects accrue over time to explain the long latency of RVCL.

We agree with the Reviewer that RVCL progression can occur in the absence of DNA damaging agents, and that disease may result from the accumulation of DNA damage over time. Now we discuss this point, in accordance with the Reviewer's feedback. Please see lines 476-478.

In the revised manuscript, we also include our new data on cytokine-induced TREX1 up-regulation leading to DNA damage in our model of RVCL. Thus, a chemotherapeutic agent is not necessary to see TREX1-associated DNA damage phenotypes in mice. Please see Figures 4 and 6.

We speculate the long latency of RVCL is also partly due to age- and cytokine-mediated up-regulation of TREX1 (in the context of inflamm-aging). Please see lines 442-446. This will be further explored as topic of future manuscripts.

Related to the point above the proposal of organ-specific TREX1 induction in aged individuals is not well-supported by the data.

In the revised manuscript, we now also include Western blots showing age-related up-regulation of TREX1 protein in the livers and brains of mice. Please see Figure 4e-f. We agree that the voyAGER dataset is limited to only showing age-related changes in RNA. This data shows age-related up-regulation of TREX1 RNA in brain and liver of humans.

In the revised manuscript, we now link the finding of age-related up-regulation of TREX1 in humans with age-related inflammation in humans and cytokine mediated up-regulation of TREX1 protein, including in mice. Please see Figures 4 and 6 and Extended Data Figure 9.

Treatment with a TREX1 small molecule inhibitor reverses DNA damage phenotypes in primary cells (Fig. 3g). However, TREX1 catalytic inhibitors have not been extensively validated leaving their potency and potential for indirect effects to be important, open questions. While I appreciate that the genetic tools used elsewhere in the manuscript are difficult to apply in this setting, I am concerned about the rigor of this particular experiment.

We agree with the Reviewer concern about validation of TREX1 inhibitors.

To mitigate the concern about specificity of inhibitors for purpose of the revised manuscript, we show that the TREX1 inhibitor has a protective effect after induction of the human RVCL mutant TREX1 in primary macrophages, and not in macrophages lacking the TREX1 mutant. Please see Reviewer Figure 2.

Additionally, we have a second manuscript in preparation showing the specificity of many newly generated TREX1 small molecular inhibitors, including a co-crystal structure of TREX1 with one of our own newly generated small molecule inhibitors, which has the same protective effect but cannot be included here because of agreements with other collaborators.

Thus, we see the same protective effects of multiple TREX1 inhibitors only when TREX1 is expressed, and with inhibitors generated independently by different groups, including an inhibitor for which we have a co-crystal structure. Please see Reviewer Figure 2.

The authors spend considerable effort to investigate the role of interferon in RVCL based on the premise that previous studies have suggested that TREX1 is caused by loss of TREX1 function. This is not a fair characterization of the field where numerous studies have demonstrated the functionality of C-terminally truncated TREX1 and the lack of IFN in RVCL (for instance see PMID: 33476576, 28420733). Even the directly cited studies (PMID: 27230542, 26320659) are taken somewhat out of context as they focus on proposed cGAS-independent roles for TREX1 including regulation of oligosaccharyltransferase activity.

Reviewer Figure 2. The TREX1 inhibitor prevents olaparib-induced DNA damage only in cells expressing RVCL mutant TREX1. Quantitation of γ H2AX expression in bone marrow-derived macrophages from RVCL mutant TREX1 (TREX1 V235Gfs)-negative and -positive mice. BMDMs were treated for 72 hours with olaparib (10 μ M) and either vehicle control or TREX1 inhibitor (10 μ M) before analysis by flow cytometry. Data represent the mean \pm SEM of n = 6 samples from 2 independent experiments, and were analyzed by one-way ANOVA with Sidák's correction. ***P < 0.001.

We apologize for this oversight. This is corrected in the revised manuscript. We now cite and discuss these key studies that demonstrated normal circulating IFN levels in RVCL^{5,6}. Please see lines 108-109 and lines 450-458 of the revised manuscript.

Reviewer #2 (Remarks to the Author):

In this study, Holley et al propose and provide evidence for a DNA damage-based theory that elucidates the mechanism underlying the clinical phenotypes of TREX1 mutations associated with retinal vasculopathy with cerebral leukoencephalopathy (RVCL). Using a range of model systems including *Drosophila*, mice and human cell culture and a variety of methods and assays, the authors show that RVCL-associated TREX1 mutants are mislocalized in the nucleus, which subsequently causes intrinsic DNA damage by disrupting homology-mediated repair processes.

The authors present key findings to support their hypothesis: 1) The Detection of DNA damage and DNA damage response in cells expressing the mutants, contingent on TREX1's nuclease activity and nuclear localization; 2) The rescue of the rough eye phenotype in *Drosophila* caused by a RVCL-associated human TREX1 mutant through the depletion of the genes involved in the DNA damage response; 3) The detection of senescence in RVCL TREX1-expressing human cells, which, again, is dependent on TREX1 nuclease activity and its nuclear localization; 4) The observation of DNA damage and cell death (mild) caused by RVCL TREX1 nuclease activity in the presence of PARPi; 5) The detection of inhibitory effects of RVCL TREX1 on HR, which also hinge on TREX1's nuclease activity and nuclear localization; 6) The observation of aneuploidy, reduced meiotic crossover and breast cancer incidents in female RVCL patients; and 7) The observation of increased sensitivity of RVCL TREX1 expressing cells to aclarubicin (although only at high concentrations) and more severe brain lesions in an RVCL patient following aclarubicin treatment. The study further shows that RVCL mutations, contrary to mutations that cause AGS and other autoimmune disorders, do not trigger interferon production in the presence or absence of exogenous cytosolic DNA stimuli, indicating a unique pathway in RVCL pathogenesis.

The results described in the manuscript provide intriguing insights into RVCL, highlighting distinct impacts of TREX1 mutations, with important implications for the treatment of RVCL. The strategic use of multiple model systems and the comprehensive examination of the RVCL TREX1's role in a range of DNA damage-related cellular processes constitute the core strengths of the study. However, to ensure rigor the study would benefit from a deeper exploration of the molecular mechanisms, and there are several concerns that need to be addressed.

We thank the Reviewer for their comments and feedback, which prompted additional work that improved our manuscript.

Points:

1. It is unclear exactly how RVCL TREX1 mutants interfere with DNA repair (especially HR) and cause DNA damage.

In our revised manuscript, we now demonstrate that RVCL mutant TREX1 causes both large and small deletions around the site of double-stranded DNA breaks, which suggests that mutant TREX1 causes end resection of 3' overhangs that are essential for homology-directed repair. Please see Figure 7.

Furthermore, in our revised manuscript, we now demonstrate that RVCL mutant TREX1 interacts directly with chromatin. We now include new mass spectrometry data revealing interactions between WT as well as mutant TREX1 and nuclear proteins. Please see Figure 8. Some of these interactions are likely indirect or via chromatin bridging.

However, TREX1 was previously reported to interact directly with PARP1, including in the absence of chromatin bridging⁷.

Our new results in the revised manuscript further suggest a strong interaction between RVCL mutant TREX1 and PARP1, as well as with other chromatin-associated proteins including XRCC5. **Please see Figure 8. We acknowledge on lines 332-333 that some of these interactions may be through chromatin bridging.**

In the revised manuscript, we also speculate about likely mechanisms, including 3' end resection inhibiting homology-directed repair, direct TREX1-mediated DNA damage, and TREX1-mediated sequestration of DNA damage repair proteins. **Please see lines 387-395.**

In this revised manuscript, we include multiple figures of new data showing specific effects on cytokine-induced damage or loss of TREX1-high cells in cell culture and in mice. **Please see Figures 4 and 6.** Thus, we now have a cytokine- and immune-related mechanism in mice. We intend to pursue further molecular mechanistic investigations as topics of future manuscripts.

2. The authors show that RVCL TREX1 expression suppressed HR and enhanced NHEJ in human cells. They also show that RAD50 depletion apparently mitigated the RVCL TREX1-induced rough eye phenotype in *Drosophila*. It is hard to reconcile these observations, considering the established role of RAD50 in facilitating HR.

We agree that the initial manuscript did not adequately address this role of Rad50.

Reviewer 1 had the same concern. **Please see our response to Reviewer 1.** The key point is that Rad50 knockdown and/or heterozygosity are protective in DNA damage phenotypes in *Drosophila*, including in the context of the rough eye phenotype.

This is in contrast to Rad50 deletion.

Please see citations^{3,4} where other groups demonstrate the protective effect of reducing Rad50 expression in DNA damage phenotypes in *Drosophila*.

Please also see lines 161-164 of the revised manuscript.

3. It is also hard to reconcile the discrepancy between the very mild PARPi sensitivity depicted in Extended Data Figure 5a and the pronounced suppression of HR repair shown in Figure 5b and 5c. The authors should compare the effects of RVCL TREX1 with BRCA1/2 deficiency on HR and PARPi sensitivity.

Mouse embryonic fibroblasts are highly heterogeneous (**Please see Extended Data Figure 5b**). The purpose of Extended Data Figure 5a of the initial submission is not to make a claim that PARP inhibitor sensitivity is high. Instead, we wanted to demonstrate one reason for elusiveness of the discovery that TREX1-high cells undergo RVCL mutant-induced DNA damage. In mouse embryonic fibroblasts, the TREX1-low cells overtake the culture (**Please see Extended Data Figure 5c**).

By contrast, in the main figures of our paper, and in conditions where TREX1 expression is high or more uniform, we clearly see effects in terms of sensitivity to DNA damage conditions.

The effect of a PARP inhibitor is large in cells that have high expression of TREX1 (BMDMs, **Please see Figure 3 of the revised manuscript**), but not in cells that have low or heterogeneous expression of TREX1 (MEFs) (**Extended Data Figure 5a**).

In the MEF experiments in Extended Data Figure 5a of the initial submission, we were intrigued by a selection event in which cells that highly express the TREX1 mutant are eliminated in favor of cells that do not highly express the TREX1 mutant. The goal of this Extended Data Figure is to underscore the subtleties of a heterogeneous population. We have re-written our discussion of MEFs treated with PARP inhibitors to increase clarity. **Please see lines 199-213 of the revised manuscript.**

In macrophages that have uniformly high TREX1 expression (e.g., after induction of TREX1 expression in response to a cytokine), we see uniform induction of DNA damage. This occurs in response to a PARP inhibitor, and also in response to cytokines, demonstrating that a PARP inhibitor is not necessary to trigger RVCL TREX1 mutant-mediated DNA damage. Please see Figure 4 of the revised manuscript.

Additionally, we attempted to study PARP inhibitor sensitivity in ID8 ovarian cancer cells expressing RVCL mutant TREX1 in the presence and absence of BRCA1/2. Unfortunately, the TREX1 mutant was so toxic when over-expressed in ID8 cells that mutant TREX1 expression was not tolerated at all.

Therefore, we conclude that RVCL mutant TREX1 is more toxic than BRCA1/2 deficiency, at least in conditions where TREX1 is highly expressed, which is a result similar to what we found in other cell types. Please see Reviewer Figure 1.

This underscores the toxicity of the TREX1 mutant, and the tremendous challenges associated with definitively characterizing the associated DNA-damage phenotypes.

4. The authors suggest an interesting hypothesis that RVCL TREX1 may digest the 3' overhang of the resected DSB ends, thereby impeding HR. They should examine whether there is a sequence loss at the HR reporter cutting site in Figure 5a in cells expressing RVCL TREX1.

We agree. In the revised manuscript, we now include next-generation sequencing around the HR reporter cut site. We found that RVCL mutant TREX1 causes increased large and small deletions around double-stranded DNA breaks, suggesting increased degradation of 3' overhangs that are necessary for efficient homology-directed repair. Please see Figure 7 of the revised manuscript.

Reviewer #3 (Remarks to the Author):

Holley et al, investigate how mislocalization of the intracellular DNase Trex1 causes small vessel disease. Trex1 is usually anchored in the ER with its catalytic center facing the cytosol. Loss of its nuclease activity causes cGAS/STING dependent systemic autoimmunity. Similar mechanisms have been suggested for c-terminal truncation mutants (lacking its membrane anchor) that cause RVCL. However, a deep understanding of the disease pathogenesis is lacking. The authors undertook a comprehensive multi-species approach to gain insights into mechanisms that could drive pathogenesis and show that DNase active Trex1 RVCL mutants interfere with DNA DSB repair. As such, RVCL TREX1 mutants sensitize to DNA damage-inducing therapies, which is contraindicated in RVCL based on the current study.

The manuscript presents extensive data that is coherent between many models and species. The study adds a lot of new information on the previously identified nuclear TREX1 with real world implications for the treatment of RVCL and cautions in DNA damage inducing therapies outside of the cancer treatment.

We thank the Reviewer for the positive assessment and helpful critiques.

I only have a few points that should be addressed:

Overexpression of WT Trex1: In Figure 1, how does human WT Trex1 cause the rough eye phenotype if it's not mislocalized as suggested by the IF-pictures throughout the manuscript? On the contrary: The quantification of IF in Figure 1D suggest that a substantial fraction of Trex1 WT is nuclear (mean ratio nuclear/cyto approx 0.75). Can the authors clarify this also in regards of using Trex1 WT overexpression in later experiments and not observing any effects? How do the expression levels of TREX1 in their systems relate to endogenous levels of TREX1 or RVCL TREX1? In Figure 2H the text says that RVCL TREX1 did not colocalize with SET, in the legend it says myc-TREX1. Was the experiment done with mutant or WT TREX1. If it was done with mutant, would their WT TREX1 colocalize with the SET complex?

We thank the Reviewer for pointing out that WT TREX1 overexpression can partially phenocopy the effect of RVCL mutant TREX1 in certain conditions. We now mention this and explain it may be an artifact of TREX1

over-expression, including some nuclear localization of WT TREX1, as the Reviewer pointed out. **Please see lines 151-152 of the revised manuscript.**

Our confocal microscopy data in multiple systems including *Drosophila* Kenyon cells, Flp-In 293T cells, and bone marrow-derived macrophages shows that most WT TREX1 is excluded from the nucleus, but a much larger fraction of RVCL mutant TREX1 is nuclear. We now mention this in the text. **Please see lines 151-155 of the revised manuscript.**

We agree that our initial submission was confounded by relying on mostly overexpression systems. Reviewer 1 had the same concern. **Please see our response to Reviewer 1.**

The main point is that our revised manuscript includes 3 new figures showing inflammation-induced up-regulation of RVCL mutant TREX1 is associated with DNA damage, even when TREX1 is expressed at the endogenous locus, under control of the endogenous promoter. Please see Figures 4, 5, and 6.

We thank the Reviewer for pointing out the mistake in the legend for Figure 2H in our initial submission. In the revised manuscript, we have clarified the Figure legend to say “myc-TREX1 (RVCL)” (**Please see Figure 2h of the revised manuscript**). Furthermore, we discuss how WT TREX1 can co-localize with the SET complex in the setting of perforin- and granzyme A-induced DNA damage⁸. **Please see lines 190-193 of the revised manuscript.**

Figure 4D: the Ifna2 levels in the HC group seems quite high. SIMOA study showed steady state levels of Ifna in the fg/ml range, here it's one log higher. Also, RVCL patients have been screened previously for IFN and showed no difference in SIMOA having Ifna in fg/ml range (PMID: 28420733). The amounts detected here, would be consistent with an SLE or AGS diagnosis in the HC group. This assay should be checked carefully.

We agree and apologize for this oversight.

Given the questions about reliability of the IFN- α 2 data in the pg/ml range, we made a decision to remove these data from the manuscript. The company insists that the levels are accurate, but we thought it better to remove this since it is not an essential point of our manuscript, especially given the availability of prior publications confirming similar pan-IFN- α expression levels in RVCL and healthy controls⁵. Please see lines 450-461 of the revised manuscript.

Blocking of HDR: The authors use a CRISPR-bases system with a ssDNA donor to study repair efficiency in cells expressing Trex1 variants and conclude, that HDR is impaired in the presence of RVCL TREX1. If I am not mistaken, this setup introduces genetic alterations via a mechanism that is termed single strand template repair involving the Fanconi anemia pathway and is distinct from HDR (PMID: 30054595). In fact, RAD50 regulates FANCD2 activity (PMID: 19609304). Can the authors provide more mechanistic insights into the exact molecular mechanism underling the repair defect and potentially adjust the manuscript accordingly? e.g. does TREX1 colocalize / compete with Faconi factors?

In the revised manuscript, we more precisely describe the homology-directed repair (HDR) assay. We thank the Reviewer for pointing this out.

Double strand break repaired by HDR using sister chromatin or exogenous double-stranded DNA plasmid as template is defined as homologous recombination (HR), and that repaired by HDR using exogenous single-stranded DNA as template is defined as single-strand template repair (SSTR). HR and SSTR are both included in HDR. Both are mediated by the same mechanism⁹. In the revised manuscript, we refer to this assay as a test of single-stranded template repair. Please see lines 303-306 and lines 380-383.

Reviewer 2 had a similar concern about the molecular mechanism of inhibition of double-strand break repair. The key point is that in the revised manuscript we show RVCL mutant TREX1 causes deletions around DNA break sites, suggesting increased end resection and loss of 3' overhangs necessary for homology-directed

repair. **Please see Figure 7 of the revised manuscript. Please see our response to a similar question from Reviewer 2.**

Additionally, we now show that RVCL mutant TREX1 interacts with chromatin. **Please see Figure 8 of the revised manuscript.**

Although we did not see a direct interaction between RVCL mutant TREX1 and Fanconi anemia factors by immunoprecipitation-mass spectrometry (**Figure 8**), we cannot rule out competitive effects with Fanconi anemia factors. We also discuss the similarities and differences between RVCL and the neurological disorder associated with Fanconi anemia mutations. More precise molecular mechanisms are a topic of ongoing and future study. **Please see lines 427-433 of the revised manuscript.**

Some of the authors published seminal papers in establishing the connection between DNA damage and activation interferon via cGAS/STING. Can the authors clarify and speculate why the DNA damage caused by TREX1 RVCL (Fig. 2 and related) is not immunogenic? Also, the authors cite work on TREX1 translocation into the nucleus (Nader et al., lead author is co-author here), which causes invasive reprogramming of cancer cells. Do the RVCL cells show similar reprogramming? This could be an alternative mechanism underlying carcinogenesis.

In the revised manuscript, we show an immunogenic effect of DNA damage caused by RVCL. We also show loss of TREX1-high cells in RVCL mutant mice in the context of cytokine-mediated TREX1 up-regulation, both in primary cultured cells and in mice. **Please see Figures 4 and 6 and Extended Data Figure 10c of the revised manuscript.**

In the revised manuscript, we also show heightened ISG induction after irradiation of MEFs heterozygous for RVCL mutant TREX1 at the endogenous locus. Furthermore, DNA damage causes IFNAR1-dependent up-regulation of TREX1, and inflammation both up-regulates TREX1 and causes increased DNA damage in cells that express RVCL mutant TREX1 at the endogenous locus. Together, these data suggest a malignant cycle of DNA damage and local inflammation, rather than systemic inflammation. **Please see Figure 4 and Extended Data Figure 10 of the revised manuscript.**

We agree that RVCL mutant TREX1 may cause reprogramming of cancer cells similar to nuclear translocation of WT TREX1 in Nader et. al¹⁰. This will be the focus of future studies.

REFERENCES

- 1 Luo, G. et al. Disruption of mRad50 causes embryonic stem cell lethality, abnormal embryonic development, and sensitivity to ionizing radiation. *Proceedings of the National Academy of Sciences* **96**, 7376-7381 (1999). <https://doi.org:10.1073/pnas.96.13.7376>
- 2 Gorski, M. M. et al. Disruption of Drosophila Rad50 causes pupal lethality, the accumulation of DNA double-strand breaks and the induction of apoptosis in third instar larvae. *DNA Repair* **3**, 603-615 (2004). <https://doi.org:https://doi.org/10.1016/j.dnarep.2004.02.001>
- 3 Rimkus, S. A. et al. Mutations in String/CDC25 inhibit cell cycle re-entry and neurodegeneration in a Drosophila model of Ataxia telangiectasia. *Genes & Development* **22**, 1205-1220 (2008). <https://doi.org:10.1101/gad.1639608>
- 4 Tuxworth, R. I. et al. Attenuating the DNA damage response to double-strand breaks restores function in models of CNS neurodegeneration. *Brain Communications* **1**, fcz005 (2019). <https://doi.org:10.1093/braincomms/fcz005>
- 5 Rodero, M. P. et al. Detection of interferon alpha protein reveals differential levels and cellular sources in disease. *Journal of Experimental Medicine* **214**, 1547-1555 (2017). <https://doi.org:10.1084/jem.20161451>
- 6 Mohr, L. et al. ER-directed TREX1 limits cGAS activation at micronuclei. *Molecular Cell* **81**, 724-738.e729 (2021). <https://doi.org:https://doi.org/10.1016/j.molcel.2020.12.037>
- 7 Miyazaki, T. et al. The 3'-5' DNA Exonuclease TREX1 Directly Interacts with Poly(ADP-ribose) Polymerase-1 (PARP1) during the DNA Damage Response*. *Journal of Biological Chemistry* **289**, 32548-32558 (2014). <https://doi.org:https://doi.org/10.1074/jbc.M114.547331>

- 8 Chowdhury, D. *et al.* The exonuclease TREX1 is in the SET complex and acts in concert with NM23-H1 to degrade DNA during granzyme A-mediated cell death. *Mol Cell* **23**, 133-142 (2006). <https://doi.org:10.1016/j.molcel.2006.06.005>
- 9 Riesenber, S. *et al.* Efficient high-precision homology-directed repair-dependent genome editing by HDRobust. *Nature Methods* **20**, 1388-1399 (2023). <https://doi.org:10.1038/s41592-023-01949-1>
- 10 Nader, G. P. d. F. *et al.* Compromised nuclear envelope integrity drives TREX1-dependent DNA damage and tumor cell invasion. *Cell* **184**, 5230-5246.e5222 (2021). <https://doi.org:https://doi.org/10.1016/j.cell.2021.08.035>

REVIEWERS' COMMENTS

Reviewer #1 (Remarks to the Author):

Thank you to the authors for enhancing the manuscript through incorporation of significant additional experimentation and for responding to each of my prior points. The majority of my prior concerns have been satisfactorily addressed.

My major concern regarding the initial manuscript was reliance on overexpression systems in key experiments. This concern has been partially addressed in the revised manuscript. The authors have moved away from the HEK293 system by generating elegant knock-in animal models that heterozygously express TREX1 RVCL mutants. Although TREX1 is now under control of its endogenous promoter, Fig. 4 uses high doses of ionizing radiation and cytokines to upregulate TREX1. Therefore, a limitation of the revised manuscript is how well these stimuli represent more physiological sources of TREX1 upregulation.

Related: Cytokine treatment causes a significant increase in DNA damage (Fig. 4i,j) even in wild-type TREX1 cells. Therefore, TREX1 RVCL mutant phenotypes may reflect interference with HR (as opposed to direct DNA damage).

Related: the authors speculate that ISG upregulation in RVCL MEFs may reflect cell-intrinsic immune activation due to DNA damage. This is speculative for the main results section as one can imagine other plausible interpretations (RVCL haploinsufficiency in terms of cGAS-STING regulation).

The authors have satisfactorily addressed my prior point regarding TREX1 full-length phenotypes and nuclear localization.

The authors have satisfactorily addressed my prior point regarding their identification of RAD50 rough eye suppression.

The authors have partially addressed my prior concern regarding PARP inhibition (see above) by incorporating additional genotoxics and cytokine treatments to upregulate TREX1.

The authors provide beautiful data to show TREX1 protein upregulation in the livers and brains of aged mice to satisfactorily address my prior point regarding the voyager dataset. They may wish to discuss how their results differ from prior analyses of TREX1 mRNA expression in peripheral blood serum across children and elderly humans (PMID: 37474626).

The specificity of TREX1 inhibitors used in this manuscript continues to be a concern. It is difficult to assess specificity/potency given the data shown.

The authors have satisfactorily addressed my prior point regarding current thinking in the field regarding the role of IFN signaling in RVCL.

Minor points

Fig. 4f appears to contain unintended brackets in the figure (in my version of the PDF).

Caught a typo 'NEHJ' in Fig. 7 legend.

Reviewer #2 (Remarks to the Author):

The authors have combined two manuscripts, and the merged paper is significantly strengthened with

the added new data. They have also addressed my questions and concerns. I would like to congratulate the authors for their comprehensive work, which has rather convincingly demonstrated that RVCL is a DNA damage syndrome.

Reviewer #3 (Remarks to the Author):

The authors have substantially expanded the manuscript addressed all my concerns and put their data into the correct context. The manuscript will be a valuable source of information for the scientific community.

Reviewer #4 (Remarks to the Author):

In this study, Chauvin et al. offer significant insights into the pathogenesis of retinal vasculopathy with cerebral leukoencephalopathy (RVCL), a dominantly inherited small vessel disease impacting multiple organs. They elucidate how RVCL-associated TREX1 variants contribute to DNA damage across humans, mice, and *Drosophila* by impairing homology-directed repair, a critical mechanism for genomic stability maintenance. This mechanistic understanding is pivotal for grasping RVCL disease pathology and related conditions. Moreover, the findings tentatively suggest broader implications for DNA damage in aging, linking aberrant TREX1 activity with premature senescence, microvascular disease, and possibly early-onset breast cancer in women with RVCL due to underlying DNA repair defects.

While the implications of these findings are promising, it is crucial to note that the evidence linking inherited C-terminal truncation variants in TREX1 to increased breast cancer odds remains preliminary. Nevertheless, this research paves the way for future investigations that may reveal a potential parallel with established predisposition genes, such as BRCA1 and BRCA2, which should not be overlooked.

The revised manuscript and rebuttal have adeptly addressed prior concerns regarding overexpression artifacts through additional mouse experiments that avoid over-expression by expressing TREX1 under its endogenous promoter. This includes new figures and data delineating cytokine interactions, DNA damage, and TREX1 expression; employing TREX1-dsRed reporter mice to identify cell types with innate high TREX1 expression; and showcasing cytokine-induced reductions in specific immune cell subsets with high TREX1 expression.

The study verifies that RVCL mutant TREX1 is deleterious at elevated expression levels, linking its cytokine-driven upregulation with DNA damage and the depletion of certain TREX1-expressing cell types. This substantiates the harmful effects of RVCL TREX1 mutants and their role in disease pathology. By integrating findings from *Drosophila*, mice, and human cells, the study presents a cohesive picture of RVCL TREX1 mutants' effects across various biological systems.

In summary, this study not only deepens our comprehension of RVCL but also introduces C-terminal truncation variants in TREX1 as a novel area of investigation in the genetic predisposition to breast cancer. This insightful connection, while exploratory, invites a necessary discourse on genetic markers beyond the well-documented BRCA genes, underscoring the importance of continued research in this domain. The additional data provided in this revised manuscript address initial concerns and substantiate the study's findings. As such, this research merits publication for its original contribution to the field and its potential to catalyze further discovery.

We thank the Reviewers for the positive assessment of our manuscript.

We have addressed all concerns by modifying the text. Please see lines 396, 407-408, 440-441, 452-453, 482-483.